# Hyperfunction of post-synaptic density protein 95 promotes seizure response in early-stage aβ pathology

Yeeun Yook [ID][1], Kwan Young Lee [ID][1], Eunyoung Kim[2], Simon Lizarazo[1], Xinzhu Yu [ID][1,2,3] & Nien-Pei Tsai [ID][1,2,3 ✉]

## Abstract

Accumulation of amyloid-beta (Aβ) can lead to the formation of aggregates that contribute to neurodegeneration in Alzheimer's disease (AD). Despite globally reduced neural activity during AD onset, recent studies have suggested that Aβ induces hyperexcitability and seizure-like activity during the early stages of the disease that ultimately exacerbate cognitive decline. However, the underlying mechanism is unknown. Here, we reveal an Aβ-induced elevation of postsynaptic density protein 95 (PSD-95) in cultured neurons in vitro and in an in vivo AD model using APP/PS1 mice at 8 weeks of age. Elevation of PSD-95 occurs as a result of reduced ubiquitination caused by Akt-dependent phosphorylation of E3 ubiquitin ligase murine-double-minute 2 (Mdm2). The elevation of PSD-95 is consistent with the facilitation of excitatory synapses and the surface expression of α-amino-3-hydroxy-5-methyl-4-isoxazolepropionic acid (AMPA) receptors induced by Aβ. Inhibition of PSD-95 corrects these Aβ-induced synaptic defects and reduces seizure activity in APP/PS1 mice. Our results demonstrate a mechanism underlying elevated seizure activity during early-stage Aβ pathology and suggest that PSD-95 could be an early biomarker and novel therapeutic target for AD.

Keywords Alzheimer's Disease; PSD-95; Seizure; Mdm2; AMPA Receptors
Subject Categories Molecular Biology of Disease; Neuroscience

## Introduction

Alzheimer's disease (AD) is a progressive neurodegenerative disease that is characterized by loss of neuronal connections and is a main cause of dementia. Approximately one in nine people who are older than 65 years is diagnosed with AD, and deaths from AD have increased by more than twofold in the past two decades. AD has two neuropathological features: extracellular accumulations of amyloid-beta (Aβ) peptides and intracellular neurofibrillary tangles (NFT), which are aggregates of hyperphosphorylated tau proteins. The abnormal accumulation of these proteins correlates with synaptic and neuronal loss, followed by brain atrophy. Multiple studies have suggested that Aβ may trigger pathological alteration that induces neuronal degeneration, while NFT facilitates Aβ-triggered neurotoxicity in later stages (Rapoport et al, 2002). This is supported by the fact that Aβ accumulation can start more than two decades before the onset of AD (Sperling et al, 2014), even before tauopathy or mild cognitive impairment. However, despite significant efforts and investments, there remains no cure for AD. Therefore, understanding how Aβ initiates neuropathology early in the disease is crucial to develop therapeutic strategies to effectively prevent irreversible damage to the brain.

Aβ comprises a group of fragments ranging from 36 to 43 amino acids in length that are generated through the proteolytic processing of amyloid precursor protein (APP) (Haass et al, 2012). Once APP is cleaved by β- and γ-secretases, Aβ is produced and released into the extracellular space as a monomer, which can subsequently aggregate into oligomers, protofibrils, fibrils, or plaques. Earlier studies revealed that insoluble amyloid plaques are the major neurotoxic factors causing cellular death (Hsiao et al, 1996; Uddin et al, 2020). However, over the past decade, the classic view has been challenged by the Aβ oligomer hypothesis, in which the oligomeric form is the more toxic and pathological agent. For example, patients who carry the Osaka (E693Δ) mutation in APP show severe cognitive malfunctions even in the absence of plaques (Bateman et al, 2012). Furthermore, the same mutant peptides impair hippocampal long-term potentiation (LTP) and induce synaptic degeneration, tau phosphorylation, microglial/astrocyte activation, neuronal loss, and memory impairment without fibrilization in rodent models (Tomiyama et al, 2010, 2008). Substantial studies with various types of AD animal models and human patients support that Aβ oligomers can instigate pathological events in AD (Cline et al, 2018; Gandy et al, 2010). These previous findings highlight the importance of studying Aβ pathology in an earlier stage.

Prior investigations primarily focused on elucidating the impact of Aβ on reduced neuronal activity, given that neuronal

[1]Department of Molecular and Integrative Physiology, School of Molecular and Cellular Biology, University of Illinois at Urbana-Champaign, Urbana, IL 61801, USA. [2]Neuroscience Program, University of Illinois at Urbana-Champaign, Urbana, IL 61801, USA. [3]Beckman Institute for Advanced Science and Technology, University of Illinois at Urbana-Champaign, Urbana, IL 61801, USA. ✉E-mail: nptsai@illinois.edu

degeneration is a distinguishing feature of AD. Nonetheless, recent evidence indicates that Aβ initially induces brain hyperexcitability before gradually leading to hypoactivity. This is supported by an observation of increased hippocampal activation measured by functional magnetic resonance imaging (fMRI) during memory-encoding tasks in individuals who carry presenilin-1 (PS1) with E280A mutation or the APOE4 allele, which are well-known genetic risk factors for AD, before the onset of clinical symptoms (Bakker et al, 2012; Bookheimer et al, 2000; Celone et al, 2006; Dickerson et al, 2005; Quiroz et al, 2010; Trivedi et al, 2008). The prevalence of seizures in AD has been found to vary from 1.5 to 64% due to methodological limitations, such as varying the sample sizes and detecting different types of seizures (Friedman et al, 2012). For example, one study showed that 42% of AD patients with no history of seizures showed subclinical epileptiform activity during 24-hour EEGs, which was four times higher than the rate in the control group (Vossel et al, 2017). Another study indicated that 28% of carriers of familial AD-related mutations in APP, PS1, or PS-2 genes developed seizures (Shea et al, 2016). In various transgenic AD animal models, neuronal hyperactivity has also been detected via two-photon $Ca^{2+}$ imaging (Busche et al, 2012, 2008) or in vivo EEG (Nygaard et al, 2015; Rudinskiy et al, 2012; Sanchez et al, 2012). In an acute Aβ model, elevated $Ca^{2+}$ transients were also observed in wild-type (WT) mice immediately after local application of Aβ onto CA1 neurons (Busche et al, 2012). AD patients with epileptiform activity experience a faster decline in cognitive functions (Vossel et al, 2016), so there is an urgent need to characterize the molecular mechanism by which Aβ induces hyperexcitability in the early stage of AD.

Postsynaptic Density Protein 95 (PSD-95) is the major scaffolding protein in postsynaptic density (PSD). PSD-95 has three PSD-95/Dlg-1/Z0-1 (PDZ) domains, a Src-homology-3 (SH3) domain, and a Guanylate kinase (GK) domain (Rademacher et al, 2019). The GK domain interacts with numerous binding partners and allows PSD-95 to interact with other scaffolding proteins, such as GKAP (Kim et al, 1997; Takeuchi et al, 1997) and AKAP 79/150 (Colledge et al, 2000). The level of PSD-95 and its activity are regulated by ubiquitination mediated by an E3 ubiquitin ligase, mouse-double-minute 2 (Mdm2) (Colledge et al, 2003). PSD-95 promotes excitatory synaptic activity by interacting with its binding partners directly or indirectly. For example, PSD-95 interacts with alpha-amino-3-hydroxy-5-methyl-4-isoxazolepropionic acid (AMPA) receptors through Stargazin, the auxiliary subunit of AMPA receptors (Bats et al, 2007; El-Husseini et al, 2000), which leads to stabilized surface expression and facilitated synaptic transmission (Schlüter et al, 2006; Schnell et al, 2002). PSD-95 also binds to GKAP and contributes to the assembly of postsynaptic scaffolding complexes with other scaffolding proteins, such as Shank and Homer (Kim et al, 1997; Naisbitt et al, 1999). This allows actin-binding proteins and metabotropic glutamate receptors to function properly via stabilized synaptic morphology. Through these mechanisms and many others, PSD-95 promotes synapse maturation (El-Husseini et al, 2000) and allows for synaptic plasticity (Béïque and Andrade, 2003). Because of these mechanisms, elevated levels of PSD-95 are associated with increased brain activity, potentially contributing to neuronal hyperexcitability and seizures.

In this study, the aim was to characterize the molecular and cellular mechanisms underlying elevated seizure activity during early Aβ pathology. We defined this stage as when animals start to

accumulate plasma Aβ but without detectable plaques or defects in memory behavior, based on a previous study (He et al, 2013). Through our research, we observed an increase of PSD-95 in cultured neurons treated with synthetic Aβ for 2 h and in a model of Aβ pathology using APP/PS1 mice at 8 weeks old. We also found that the elevation is attributed to reduced interaction with Mdm2, which leads to a decrease in the ubiquitination of PSD-95. Second, to assess the impact of elevated PSD-95, we conducted unbiased proteomic profiling of PSD-enriched fractions and discovered elevated levels of multiple synaptic proteins in the PSD of APP/PS1 mice, including Glutamate receptor 1 (GluA1), one of the major subunits of AMPA receptors. Consistent with this observation, our immunocytochemistry data showed that treatment of Aβ in cultured neurons can increase the number of synapses and the surface expression of AMPA receptors. Most importantly, we showed that genetically inhibiting PSD-95 can ameliorate these synaptic alterations and significantly reduce the seizure response in APP/PS1 mice. In summary, our study provides the first demonstration of the molecular mechanisms underlying elevated seizure response during the early stage of Aβ pathology and introduces PSD-95 as an early biomarker and potential therapeutic target for alleviating seizure attacks in AD.

## Results

### Young APP/PS1 mice exhibit higher response to kainic acid-induced seizures

Studies have observed non-convulsive seizure activity and epileptiform discharges in AD patients and Aβ transgenic mouse models during or after the onset of clinical symptoms (Minkeviciene et al, 2009; Palop et al, 2007; Palop and Mucke, 2010; Um et al, 2012). However, we do not know the response to seizures during the earlier stage of Aβ pathology. To address this question, we employed a kainic acid (KA)-induced seizure model in a well-established mouse model of Aβ pathology, double-transgenic APP/PS1 mice, at 8 weeks of age. We chose this age because we aim to understand the early effects of Aβ before the formation of plaques that were typically observed in these mice between 4 and 6 months of age (Jackson et al, 2013; Minkeviciene et al, 2008). In this experiment, APP/PS1 mice and their WT littermates were habituated for 30 min in a testing cage, followed by a single intraperitoneal injection of KA (15 mg/kg). This dosage was chosen based on our previous studies, which showed consistent and relatively mild seizures in WT mice of a C57BL/6J background (Jewett et al, 2016; Zhu et al, 2017). After the injection, the seizures were closely monitored for 1 h in real time (Fig. 1A). The intensity of behavioral seizures was measured by a modified Racine scoring scale (Fig. 1B). As we have done previously (Liu et al, 2021), only stages 4 and 5 based on this scale were considered positive for seizures, as stages 3 and below are often difficult to characterize. As shown (Fig. 1C), compared to WT littermate controls, APP/PS1 mice exhibited significantly higher susceptibility to seizures ($1C_1$) and seizure intensity ($1C_2$). Furthermore, the lethality following seizures was also higher in APP/PS1 mice ($1C_3$). Although other studies have shown that female APP/PS1 mice exhibit higher seizure susceptibility after 12 weeks of age (Minkeviciene et al,

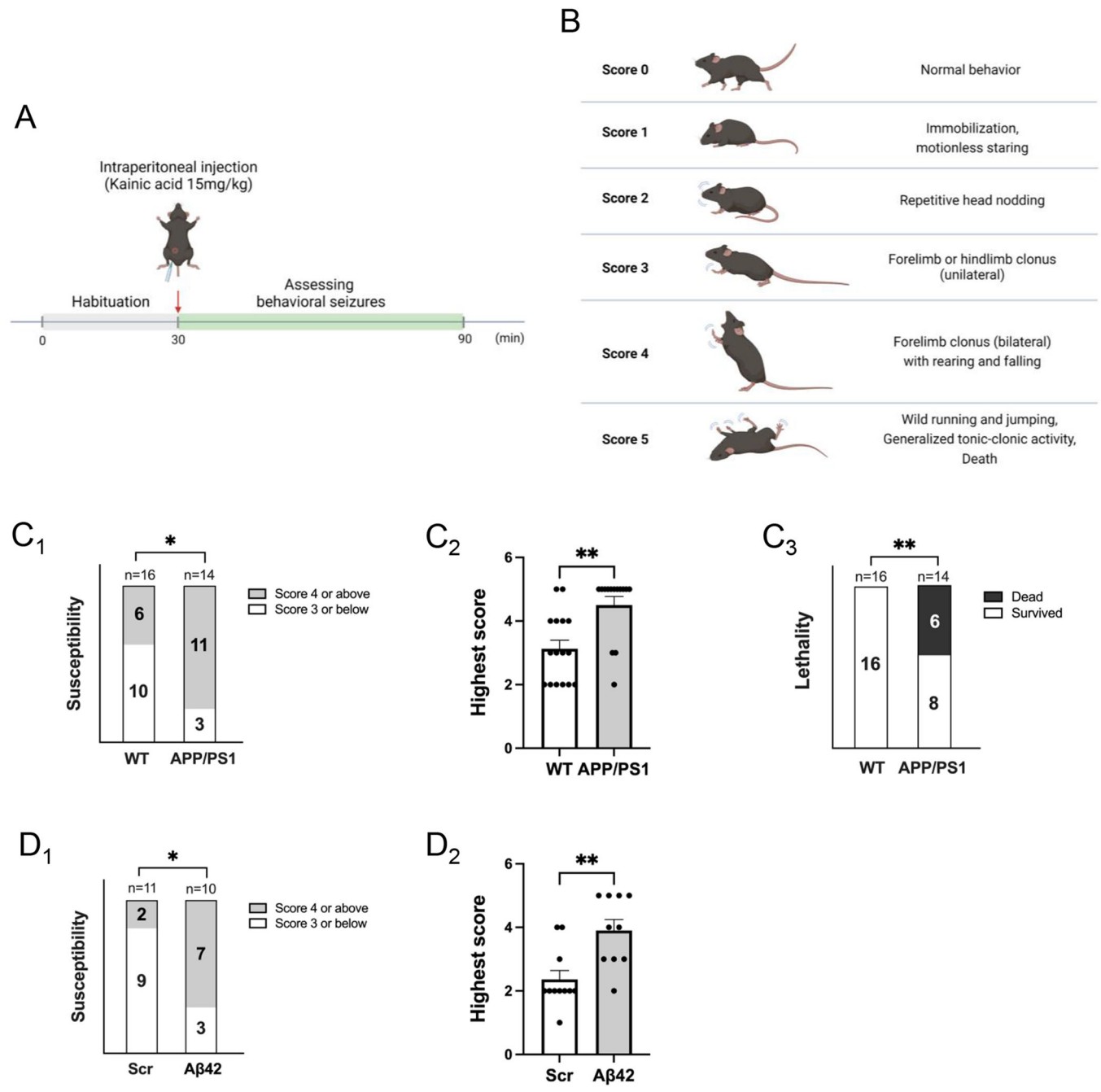

**Figure 1. Young APP/PS1 mice exhibit higher response to kainic acid-induced seizures.**

(A) A schematic of seizure observation following intraperitoneal injections of kainic acid. (B) A summary chart of the modified Racine scoring scale for evaluation of seizures. (C) Quantification of susceptibility to stage 4 seizures (*P = 0.0329) (C1), highest seizure scores (**P = 0.0013) (C2) and lethality (**P = 0.0051) (C3) from 8-weeks-old APP/PS1 mice or their WT littermates intraperitoneally injected with kainic acid (15 mg/kg). n = 16 and 14 for WT and APP/PS1 mice, respectively. (D) Quantification of susceptibility to stage 4 seizures (*P = 0.03) (D1) and highest seizure scores (**P = 0.0034) (D2) from 8-weeks-old WT mice injected with amyloid-beta 1–42 (Aβ42) or scrambled peptide (Scr). Mice were intraperitoneally injected with kainic acid (15 mg/kg). n = 16 and 14 for WT and APP/PS1 mice, respectively. Data Information: significance was determined by Fisher's exact test (seizure susceptibility and lethality) or Mann–Whitney U test (highest score). Data are represented as mean ± SEM with *P < 0.05 and **P < 0.01. Source data are available online for this figure.

2009; Reyes-Marin and Nuñez, 2017), we did not observe sex differences in our mice at 8 weeks of age (Fig. EV1). This suggests that sex differences may start appearing between 8 and 12 weeks of age in APP/PS1 mice.

It is known that multiple peptide products, in addition to Aβ$_{1-42}$, are elevated as a result of elevated APP processing in APP/PS1 mice (Szögi et al, 2022). To establish the causality between Aβ$_{1-42}$ and seizure response, we unilaterally injected synthetic Aβ$_{1-42}$ peptides

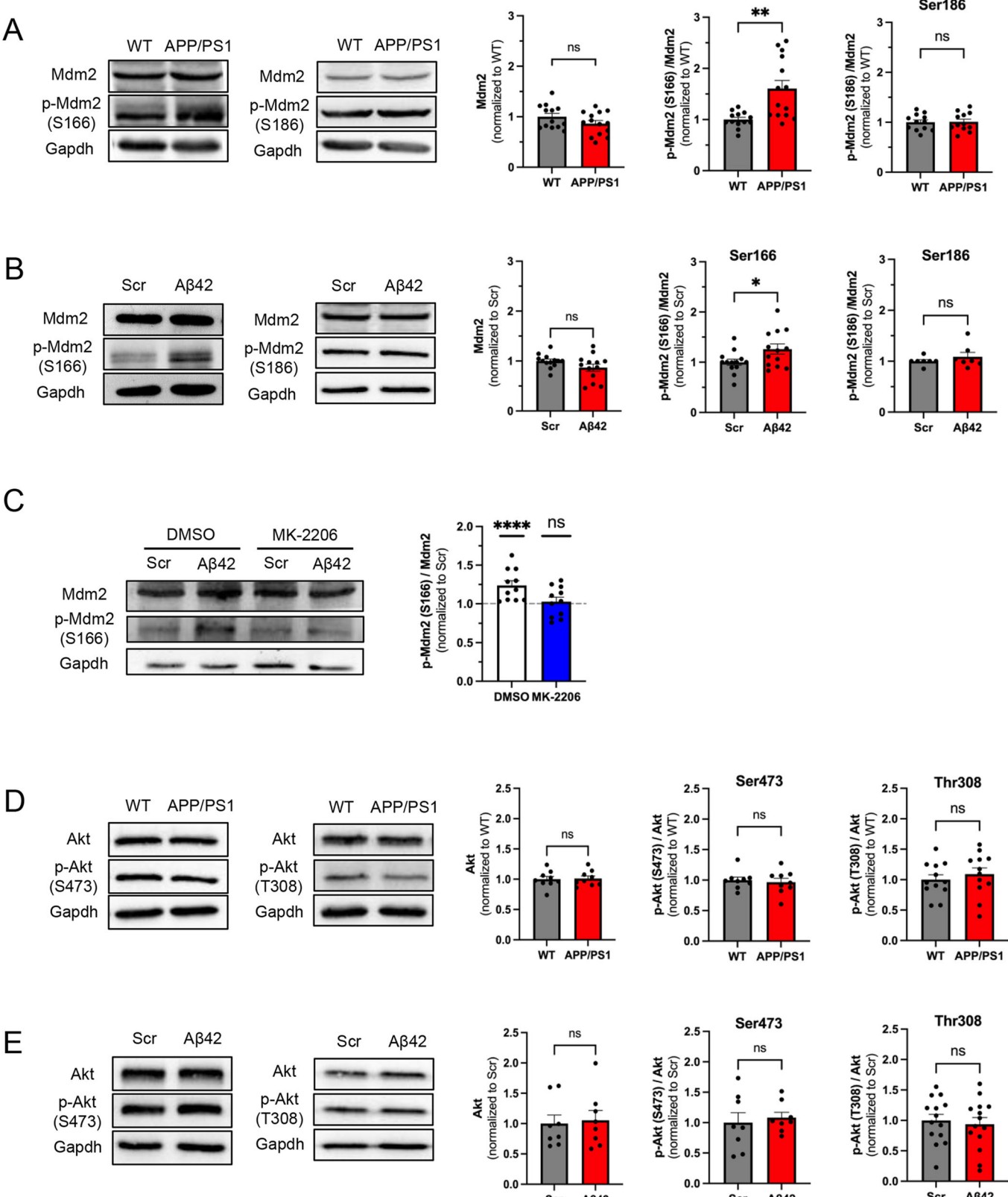

**Figure 2. Aβ promotes Akt-dependent Mdm2 phosphorylation in vitro and in vivo.**

(A) Representative western blots from total brain lysate of APP/PS1 mice or their WT littermates at 8 weeks of age (left). Quantification of Mdm2 ($P = 0.1442$), phosphor (p)-Mdm2 at S166 (**$P = 0.0019$), and p-Mdm2 at S186 ($P = 0.9126$) (right). $n = 12$–14 for WT and APP/PS1 mice, respectively. (B) Representative western blots from WT primary cortical neuron cultures treated with amyloid-beta 1–42 (Aβ42; 1 μM) or scrambled peptide (Scr, 1 μM) for 2 h at DIV 12–14. (Left) Quantification of Mdm2 ($P = 0.1169$), phosphor (p)-Mdm2 at S166 (*$P = 0.0363$), and p-Mdm2 at S186 ($P = 0.6125$) (right). For Mdm2 and p-Mdm2 at S166, $n = 13$ from seven independent cultures for both Scr and Aβ42 groups. For p-Mdm2 at S186 $n = 7$ from three independent cultures. (C) Representative western blots from WT primary cortical neuron cultures treated with MK-2206 for 30 min, followed by Aβ42 or Scr (1 μM) for 2 h at DIV 12–14 (left). Quantification is performed by first normalizing p-Mdm2 at S166 to total Mdm2 levels, followed by normalizing Aβ42 group to Scr group within DMSO or MK-2206 group. $n = 11$ from six independent cultures for both Scr and Aβ42 groups. (D) Representative western blots from total brain lysate of APP/PS1 mice or their WT littermates at 8 weeks of age (left). Quantification of Akt ($P = 0.8666$), p-Akt at S473 ($P = 0.6823$), and p-Akt at T308 ($P = 0.4952$) (right). For Akt and p-Akt at S473, $n = 9$ for WT and APP/PS1 mice. For p-Akt at T308, $n = 12$ for WT and APP/PS1 mice. (E) Representative western blots from WT primary cortical neuron cultures treated with Aβ42 or Scr (1 μM) for 2 h at DIV 12–14 (left) Quantification of Akt ($P = 0.8785$), p-Akt at S473 ($P = 0.6669$), and p-Akt at T308 ($P = 0.6716$) (right). For Akt and p-Akt at S473, $n = 8$ from two independent cultures for Scr and Aβ42 groups. For p-Akt at T308, $n = 14$ from three independent cultures for Scr and Aβ42 groups. Data Information: Significance was determined by Student's $t$ test (D) or Mann–Whitney $U$ test (A, B, C, E). Data are represented as mean ± SEM with *$P < 0.05$, **$P < 0.01$, ****$P < 0.0001$, ns non-significant. Source data are available online for this figure.

or a scrambled (Scr) peptide (1.25 μM, 4 μl) into the ventricle of WT mice at 7 weeks of age. The concentration and volume of peptides were chosen based on a previous study (Schmid et al, 2017). Five to 6 days after the injections, mice were intraperitoneally administered with KA (15 mg/kg) and monitored for an hour. As shown, mice injected with Aβ$_{1-42}$ showed higher seizure susceptibility (Fig. 1D$_1$) and seizure severity (Fig. 1D$_2$) compared to those injected with the scrambled peptide. No lethality was observed in either group, suggesting the overall burdens following seizure is likely not as high as those in APP/PS1 mice. Together, our data suggest that Aβ leads to elevated seizure response.

## Aβ promotes Mdm2 phosphorylation in vitro and in vivo

To begin characterizing the molecular mechanism underlying elevated seizure susceptibility in young APP/PS1 mice, we studied Mdm2, an E3 ubiquitin ligase with known functions in regulating seizures and neural network activity (Jewett et al, 2016; Liu et al, 2019, 2017). Because Mdm2 phosphorylation at serine 166 (S166) and serine 186 (S186) modulates the activity of Mdm2 (Chen, 2012; Chibaya et al, 2021; Liu et al, 2019), we asked whether Aβ could alter the protein levels or the phosphorylation status of Mdm2. As shown in Fig. 2A, while the total level of Mdm2 and its phosphorylation at S186 were not altered, phosphorylation at S166 was significantly elevated in total brain lysates of APP/PS1 mice. To determine whether our observation was specific to the elevation of Aβ, we took an acute approach and incubated WT cortical neuron cultures at 12–14 days in vitro (DIV) with Aβ$_{1-42}$ peptide (1 μM) or a scrambled peptide (control; 1 μM) for 2 h, which is a duration that was employed previously (Lapresa et al, 2019). We chose 1 μM based on previous studies showing successful effects on neuronal hyperactivity (Brorson et al, 1995; Ciccone et al, 2019). As shown in Fig. 2B, Mdm2 phosphorylation at S166 was upregulated in cultures treated with Aβ$_{1-42}$ in comparison to those treated with the scrambled peptide. No changes were observed in total protein levels of Mdm2 or Mdm2 phosphorylation at S186. These data suggest that Aβ induces elevation of Mdm2 phosphorylation at S166 in vitro and in vivo.

To explore the mechanism underlying Aβ-induced Mdm2 phosphorylation, we first studied eukaryotic translation elongation factor-1-alpha (eEF1α). We previously showed that elevated interaction between Mdm2 and eEF1α can interfere with the interaction between Mdm2 and protein phosphatase 2A (PP2A),

leading to elevated phosphorylation of Mdm2 (Tsai et al, 2016). To test whether eEF1α is involved in Aβ-induced Mdm2 phosphorylation, we performed co-immunoprecipitation (co-IP) using anti-Mdm2 antibody and western blotting using anti-eEF1α antibody in WT cortical neuron cultures treated with Aβ$_{1-42}$. To quantify, the precipitated eEF1α is normalized to the precipitated Mdm2. However, we did not observe changes in their interaction (Fig. EV2). We next asked whether Aβ induces Mdm2 phosphorylation depending on Akt, the major kinase that phosphorylates Mdm2 at S166 (Chen, 2012; Mayo and Donner, 2001). To answer this question, primary cortical neuron cultures were pre-treated with an Akt inhibitor MK-2206 (5 μM) for 30 mins before the treatment with Aβ$_{1-42}$ peptide or scrambled peptide (1 μM) for 2 h. As shown (Fig. 2C), Aβ-induced Mdm2 phosphorylation at S166 was disrupted in MK-2206-treated neurons, confirming the role of Akt in this process. Because Akt activity can be facilitated by its phosphorylation at serine 473 (S473) or threonine 308 (T308) residues (Manning and Toker, 2017), we tested whether Aβ promotes Akt phosphorylation on these residues. As shown (Fig. 2D,E), however, we did not observe significant changes in Akt phosphorylation at either of these residues. Altogether, our data suggest that Aβ-induced Mdm2 phosphorylation is mediated by Akt but is likely independent of Akt phosphorylation.

## Aβ-induced Mdm2 phosphorylation leads to an elevation of PSD-95 via reduced ubiquitination

Phosphorylation of Mdm2 at S166 promotes its nuclear distribution and subsequently the binding to its nuclear substrates, such as the tumor suppressor p53 (Liu et al, 2019; Mayo and Donner, 2001). To determine whether Aβ causes changes in the levels of p53, we performed western blotting, which showed no significant changes in the levels of p53 in vivo in 8-week-old APP/PS1 mice (Fig. 3A) or in vitro in WT cortical neuron cultures treated with Aβ$_{1-42}$ for 2 h (Fig. 3B). These data suggest that Aβ-induced Mdm2 phosphorylation might alter the levels of other substrates. Previous studies have demonstrated Mdm2-mediated ubiquitination and degradation of postsynaptic density protein PSD-95 (Bianchetta et al, 2011; Colledge et al, 2003; Tsai et al, 2016, 2012). Because we have shown that phosphorylation of Mdm2 leads to reduced ubiquitination of PSD-95 (Tsai et al, 2016, 2012), we speculate that Aβ-induced Mdm2 phosphorylation may lead to an elevation of PSD-95. To test this possibility, we first performed western blotting

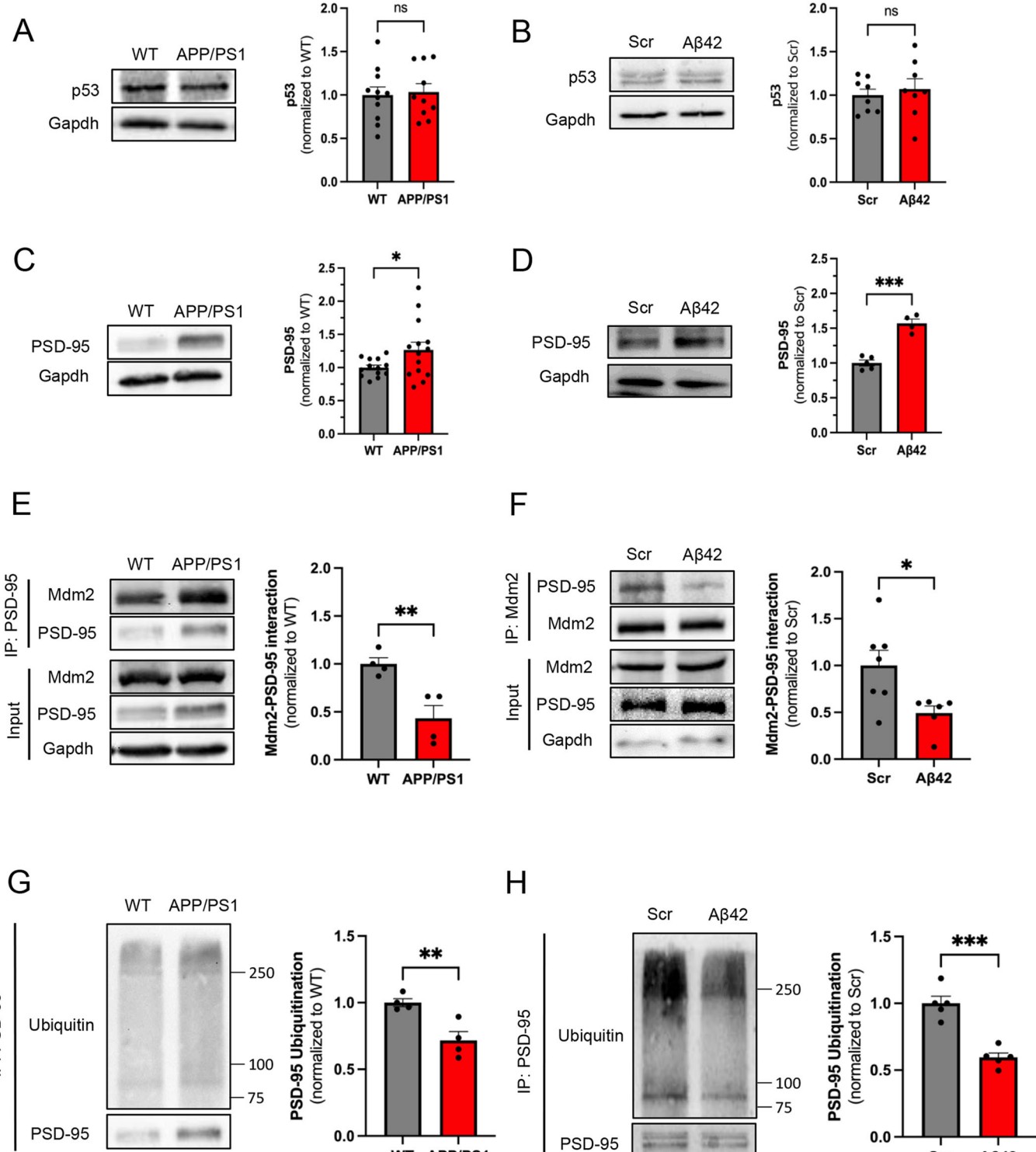

and confirmed an elevation of PSD-95 in vivo in 8-week-old APP/PS1 mice (Fig. 3C) and in vitro in WT cortical neuron cultures treated with Aβ$_{1-42}$ (Fig. 3D). Next, we asked whether the elevated PSD-95 is resulting from a reduction in Mdm2-mediated ubiquitination. To this end, we first performed co-IP using anti-PSD-95 antibody and western blotting using anti-PSD-95

and anti-Mdm2 antibodies. To quantify, the precipitated Mdm2 is normalized to the precipitated PSD-95. As shown, we observed reduced interaction between PSD-95 and Mdm2 in the lysates of APP/PS1 brain (Fig. 3E) as well as the lysates of Aβ$_{1-42}$-treated WT neuronal cultures (Fig. 3F). We next assess PSD-95 ubiquitination by immunoprecipitating PSD-95 and western blotting with

**Figure 3. Aβ-induced Mdm2 phosphorylation leads to an elevation of PSD-95 via reduced ubiquitination.**

(A) Quantification of p53 ($P = 0.7976$) and representative western blots from total brain lysate of APP/PS1 mice or their WT littermates at 8 weeks of age. $n = 11$ and 10 for WT and APP/PS1 mice, respectively. (B) Quantification of p53 ($P = 0.6118$) and representative western blots from WT primary cortical neuron cultures treated with amyloid-beta 1–42 (Aβ42; 1 μM) or scrambled peptide (Scr, 1 μM) for 2 h at DIV 12–14. $n = 8$ from two independent cultures for Ctrl and Aβ42 groups. (C) Quantification of PSD-95 (*$P = 0.0477$) and representative western blots from total brain lysate of APP/PS1 mice or their WT littermates at 8 weeks of age. $n = 13$ and 14 for WT and APP/PS1 mice, respectively. (D) Quantification of PSD-95 (***$P = 0.0001$) and representative western blots from WT primary cortical neuron cultures treated with Aβ42 (1 μM) or Scr (1 μM) for 2 h at DIV 12–14. $n = 5$ and 4 from three independent cultures for Scr and Aβ42 groups, respectively. (E) Quantification of interaction between PSD-95 and Mdm2 ($P = 0.0084$) after co-immunoprecipitation and representative western blots using lysates from APP/PS1 mice or their WT littermates at 8 weeks of age. $n = 4$ mice per genotype. (F) Quantification of interaction between PSD-95 and Mdm2 (*$P = 0.0221$) after co-immunoprecipitation and representative western blots using lysates from WT primary cortical neuron cultures treated with Aβ42 or Scr (1 μM) for 2 h at DIV 12–14. $n = 7$ and 6 from two independent cultures for Scr and Aβ42 groups, respectively. (G) Representative western blots of Ubiquitin and PSD-95 after immunoprecipitation with anti-PSD-95 antibody using lysates from APP/PS1 mice or their WT littermates at 8 weeks of age. Quantification is performed by first normalizing ubiquitinated PSD-95 (IP: PSD-95, IB: Ub) to immunoprecipitated PSD-95 (IP: PSD-95, IB: PSD-95), followed by normalizing APP/PS1 group to WT group (**$P = 0.0081$). $n = 4$ mice per genotype. (H) Representative western blots of Ubiquitin and PSD-95 after immunoprecipitation with anti-PSD-95 antibody using lysates from WT primary cortical neuron cultures treated with Aβ42 or Scr (1 μM) for 2 h at DIV 12–14. Quantification is performed by first normalizing ubiquitinated PSD-95 (IP: PSD-95, IB: Ub) to immunoprecipitated PSD-95 (IP: PSD-95, IB:PSD-95), followed by normalizing Aβ42 group to Scr group (***$P = 0.0002$). $n = 5$ from three independent cultures for both groups. Data Information: Significance was determined by Student's $t$ test (A–E, G, H) or Mann–Whitney $U$ test (F). Data are represented as mean ± SEM with *$P < 0.05$, **$P < 0.01$, ***$P < 0.001$, ns non-significant. Source data are available online for this figure.

anti-ubiquitin antibody. To quantify, we measured the ubiquitin signal above 75 kDa and normalized it to the precipitated PSD-95 signal. As shown, we observed reduced PSD-95 ubiquitination in lysates of APP/PS1 brain (Fig. 3G) as well as the lysates of Aβ$_{1-42}$-treated WT neuronal cultures (Fig. 3H). These results suggest that Aβ leads to an elevation of PSD-95 potentially via elevated Mdm2 phosphorylation and reduced PSD-95 ubiquitination.

## Aβ-induced elevation of PSD-95 leads to enrichment of a variety of postsynaptic proteins at PSD

PSD-95 can promote synaptic strength by stabilizing postsynaptic protein complexes. Following the observation of elevated PSD-95 induced by Aβ, we examined whether and how Aβ alters the participation of PSD-associated proteins in PSD. To this end, we obtained PSD fractions of cortices from WT or APP/PS1 mice at 8 weeks of age according to a published protocol (Jang et al, 2017) (Fig. 4A, left). We validated the enrichment of PSD using samples from the cytosol fractions (S2), synaptosome fraction (P2), non-PSD fraction, and finally the PSD fraction (Fig. 4A, right). We then performed label-free proteomic profiling to compare the composition of PSD fractions between WT and APP/PS1 mice. Because of potential variability between samples during fractionation, data normalization is needed in order to quantitatively compare between samples. However, there is no information thus far indicating which PSD protein might be unchanged in APP/PS1 mice that can serve as an internal control. Since elevated PSD-95 in APP/PS1 mice likely have bigger or more condensed PSD in the PSD fractions, normalizing the levels of an individual protein in PSD to a random PSD protein would not be appropriate. Therefore, we chose to normalize the levels of each protein to the levels of PSD-95. Although any proteins whose elevation is smaller than that of PSD-95 may be overlooked, which is a limitation, normalization through PSD-95 can reveal the proteins that are being further enriched in the PSD fractions of APP/PS1 mice, even beyond the already enriched PSD-95 (Dataset EV1).

As shown in Fig. 4B, we defined proteins that exhibited a log-fold change that was greater than 0 as being upregulated while those that exhibited a log-fold change that was less than 0 as being downregulated. As a result, from those with a $P$ value less than 0.05,

657 proteins were identified as upregulated, 21 proteins were downregulated, and 1439 proteins as unchanged (Fig. 4B). The volcano plot showed that PSD-95 elevation positively contributed to the recruitment of many proteins to PSD. When searching for known binding partners of PSD-95 or known regulators of post-synapses, we observed a significant elevation of many proteins in such categories, including Glutamate receptor 1 (GluA1), scaffolding proteins (Homer1/2 and Shank3), cytoskeletal proteins (Map1lc3b, Map4, Map6, Mapt, and Cofilin1), calcium voltage-gated channel (Cacnb1, Cacna2d1), and cell adhesion proteins (Neuroligin3, Ncam1, and Neuroplastin) (Fig. 4C). These findings suggest that Aβ might elevate seizure activity by recruiting these proteins to promote and strengthen PSD.

## Aβ induces PSD-95-dependent elevation of synapse numbers and surface expression of AMPA receptors

Our proteomic profiling data suggest a possibility that Aβ might promote synaptic strengths via elevation of PSD-95. To test this, we first measured the synapse numbers in WT primary cortical neurons upon treatment with Aβ$_{1-42}$ or the scrambled peptide for 2 h. Neurons were fixed, permeabilized, and immunostained with Synapsin-I, PSD-95, and MAP2 to label presynaptic, postsynaptic, and dendritic compartments, respectively. The number of synapses was measured by quantifying the number of colocalized pre- and postsynaptic puncta, as we performed previously (Lee et al, 2021; Lizarazo et al, 2022). As shown in Fig. 5A, the number of synaptic puncta was significantly enhanced in Aβ$_{1-42}$-treated neurons in comparison to scrambled peptide-treated neurons. When analyzing the number of Synapsin-I puncta and PSD-95 puncta separately, we also observed their elevations in Aβ$_{1-42}$-treated neurons. The total levels of Synapsin-I were not changed after Aβ$_{1-42}$ peptide treatment (Fig. EV3), suggesting that Aβ$_{1-42}$ may promote the recruitment of existing presynaptic compartments. Next, to determine whether elevated PSD-95 contributes to these effects, we obtained PSD-95 heterozygous (PSD-95$^{+/-}$) mice and performed the same Aβ$_{1-42}$ peptide treatment and immunostaining using cultured PSD-95$^{+/-}$ cortical neurons. As shown in Fig. 5B, knocking down PSD-95 in PSD-95$^{+/-}$ neurons blunted the Aβ$_{1-42}$-induced elevation of synaptic puncta as well as PSD-95 and Synapsin-I puncta.

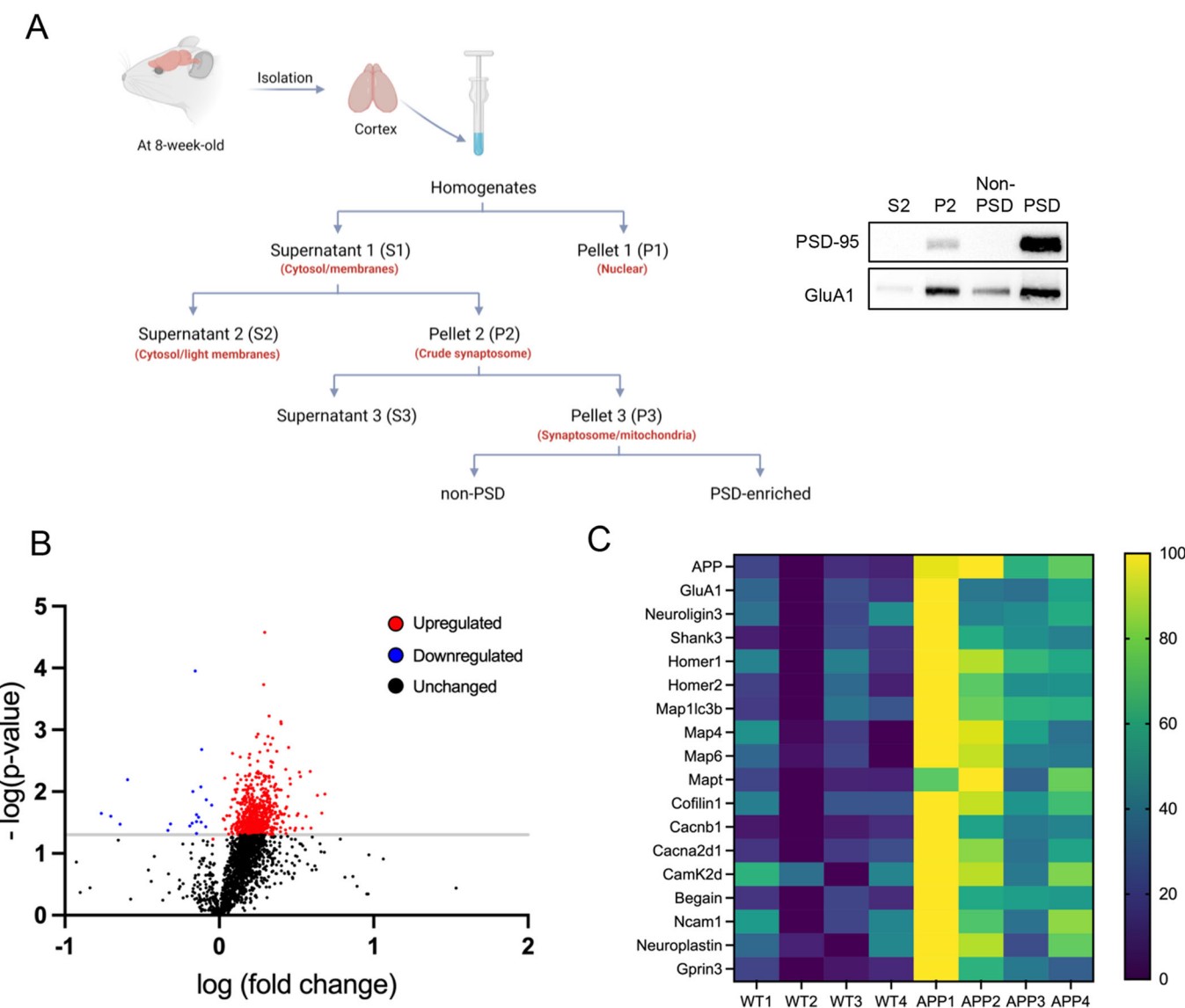

**Figure 4. Aβ-induced elevation of PSD-95 leads to enrichment of a variety of postsynaptic proteins at PSD.**

(A) A schematic of PSD fractionation (top) and an example of western blots demonstrating the results of PSD enrichment (bottom). (B) A volcano plot following RNA sequencing analysis using PSD fractions from four APP/PS1 mice and four WT littermates at 8 weeks of age. The raw value of each gene is normalized to the levels of PSD-95 from the same sample and Student's *t* test was used to determine the *P* value of each gene between WT and APP/PS1 mice where differences are considered significant when *P* < 0.05. (C) A heatmap showing 18 significantly elevated postsynaptic or postsynapse-related proteins in PSD fractions from APP/PS1 mice. Source data are available online for this figure.

To strengthen the role of PSD-95 in Aβ$_{1-42}$-induced elevation of synaptic puncta, we employed cultures prepared from PSD-95 knockout (*PSD-95$^{-/-}$*) mice. Using Homer1b/c and Synapsin-I as postsynaptic and presynaptic markers, respectively, we found that Aβ$_{1-42}$-induced elevation of colocalized pre- and postsynaptic puncta as well as Synapsin-I puncta and Homer1b/c puncta is absent in *PSD-95$^{-/-}$* neurons receiving a control lentivirus (Fig. 5C) but can be restored after lentivirally re-introducing *PSD-95* for 5 days (Fig. 5D). Together, these results confirmed that PSD-95 is required for Aβ$_{1-42}$-induced elevation of excitatory synapses in cultured neurons. Because PSD-95$^{-/+}$ and PSD-95$^{-/-}$ behave similarly in this experiment, we chose PSD-95$^{-/+}$ for our experiments hereafter.

PSD-95 promotes the strength of excitatory synapses primarily by sustaining the surface expression of AMPA receptors, and our proteomic profiling identified the GluA1 subunit of AMPA receptors as one of the significantly elevated molecules in the PSD fraction of APP/PS1 mice (Fig. 4C). Therefore, we examined whether Aβ can promote the surface expression of AMPA receptors, and if so, whether PSD-95 plays a role. To this end, we measured the surface levels of GluA1 and GluA2 in WT or *PSD-95$^{+/-}$* cortical neuron cultures treated with Aβ$_{1-42}$ or the scramble peptide for 2 h. GluA1 and GluA2 are the two most prominent subunits of synaptic AMPA receptors in the cortex (Henley and Wilkinson, 2016), and both consist of an extracellular amino-

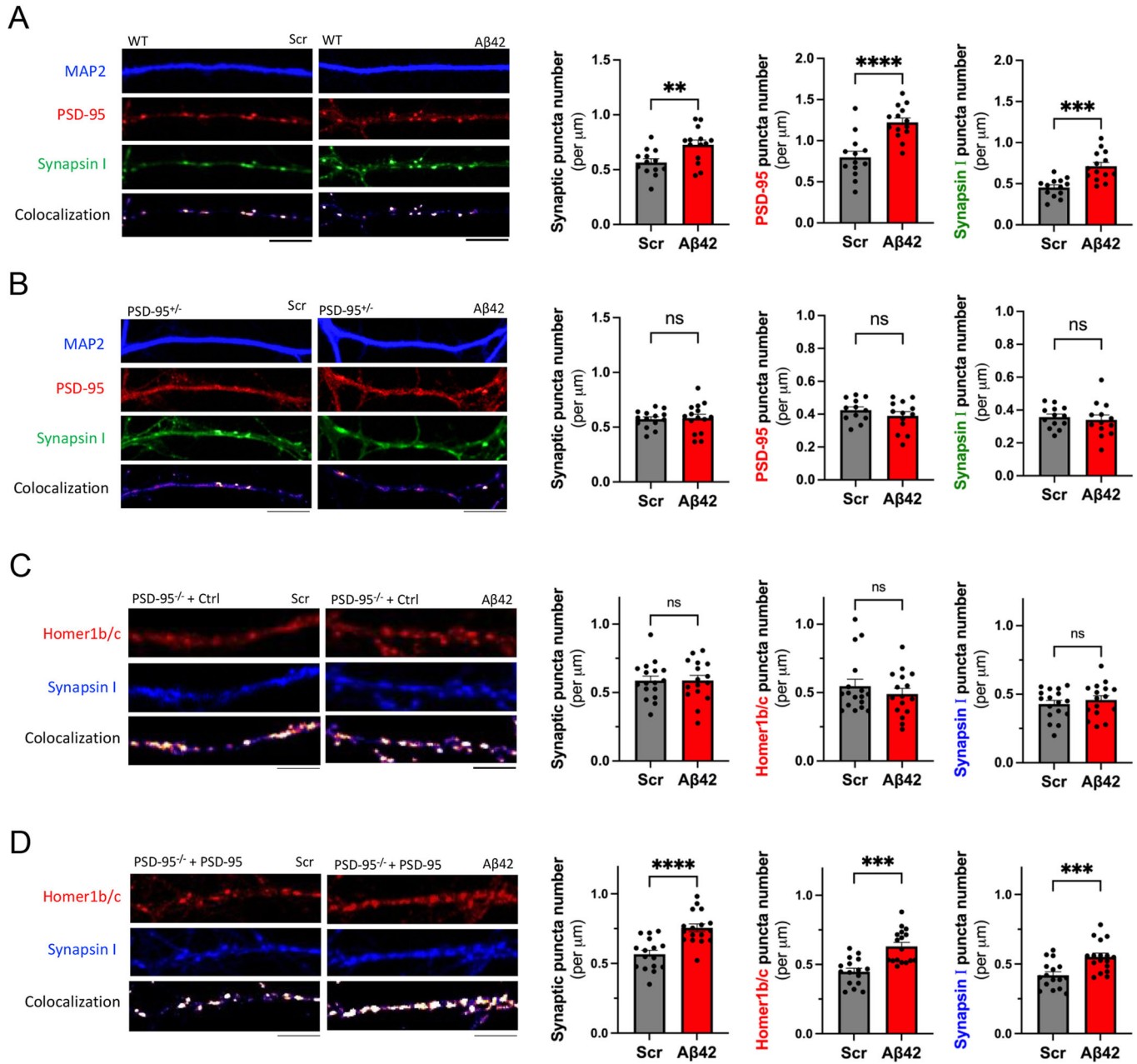

**Figure 5.  Aβ induces PSD-95-dependent elevation of synapse numbers.**

(A) Immunocytochemistry from WT primary cortical neuron cultures treated with Aβ$_{1-42}$ (Aβ42, 1 μM) or scrambled Aβ peptide (Scr, 1 μM) for 2 h at DIV 12–14 showing postsynaptic marker PSD-95 (red), presynaptic marker Synapsin-I (green), dendritic marker MAP2 (blue), and colocalization of PSD-95 and Synapsin-I (left). Quantification of colocalized synaptic puncta number (**$P = 0.0069$), PSD-95 puncta number (****$P < 0.0001$), and Synapsin-I puncta number (***$P = 0.0001$) (right). $n = 14$ cells from two independent cultures treated with Scr or Aβ42. (B) Immunocytochemistry from PSD-95$^{+/-}$ (Het) primary cortical neuron cultures treated with Aβ42 (1 μM) or Scr (1 μM) for 2 h at DIV 12–14 showing postsynaptic marker PSD-95 (red), presynaptic marker Synapsin-I (green), dendritic marker MAP2 (blue), and colocalization of PSD-95 and Synapsin-I (left). Quantification of colocalized synaptic puncta number ($P = 0.8249$), PSD-95 puncta number ($P = 0.302$), and Synapsin-I puncta number ($P = 0.6488$) (right). $n = 12$–15 cells from two independent cultures treated with Scr or Aβ42. (C) Immunocytochemistry from PSD-95$^{-/-}$ (KO) primary cortical neuron cultures transduced with control lentivirus at DIV 9–10 and treated with Aβ42 or Scr (1 μM) for 2 h at DIV 12–14 showing postsynaptic marker Homer1b/c (red), presynaptic marker Synapsin-I (green), and colocalization of Homer1b/c and Synapsin-I (left). Quantification of colocalized synaptic puncta number ($P = 0.9962$), Homer1b/c puncta number ($P = 0.5814$), and Synapsin-I puncta number ($P = 0.4667$) (right). $n = 16$–17 cells from two independent cultures treated with Scr or Aβ42. (D) Immunocytochemistry from PSD-95$^{-/-}$ (KO) primary cortical neuron cultures transduced with *PSD-95* lentivirus at DIV 9–10 and treated with Aβ42 or Scr (1 μM) for 2 h at DIV 12–14 showing postsynaptic marker Homer1b/c (red), presynaptic marker Synapsin-I (green), and colocalization of Homer1b/c and Synapsin-I (left). Quantification of colocalized synaptic puncta number (****$P < 0.0001$) Homer1b/c puncta number (***$P = 0.0001$), and Synapsin-I puncta number (***$P = 0.0009$) (right). $n = 16$–17 cells from two independent cultures treated with Scr or Aβ42. Data Information: significance was determined by Student's *t* test (A–C) or Mann–Whitney *U* test (D). Data are represented as mean ± SEM with *$P < 0.05$, **$P < 0.01$, ***$P < 0.001$, ****$P < 0.0001$, ns: non-significant. Scale bar: 5 μm. Source data are available online for this figure.

terminal (N-terminal) domain and an intracellular carboxy-terminal (C-terminal) domain (Traynelis et al, 2010). To immuno-label surface GluA1 or GluA2, we employed antibodies that recognize the N-terminus of GluA1 or GluA2 before permeabilization. The same cultures were then immunostained with antibodies that recognize intracellular the C-terminus of GluA1 or GluA2 after permeabilization to detect the total levels of GluA1 and GluA2. The ratio of surface to total expression of GluA1 or GluA2 was quantified, and the data indicated a significant elevation of the surface expression of both GluA1 and GluA2 following $A\beta_{1-42}$ peptide treatment in WT neurons (Fig. 6A,B). As expected, these effects were inhibited in $PSD-95^{+/-}$ neurons (Fig. 6C,D). Altogether, our results suggest that $A\beta$ promotes surface expression of AMPA receptors and that the effect depends on the availability of PSD-95.

### PSD-95 suppression significantly reduces seizure severity in APP/PS1 mice

Our findings thus far suggest that $A\beta$ elevates the levels of PSD-95 and triggers PSD-95-dependent facilitation of excitatory synapses and surface expression of AMPA receptors. These observations prompted us to examine whether PSD-95 contributes to elevated seizure susceptibility in young APP/PS1 mice. To this end, we crossed APP/PS1 mice with PSD-95 heterozygous mice ($PSD-95^{+/-}$, Fig. 7A) to generate four experimental groups of littermate mice: WT, $PSD-95^{+/-}$, APP/PS1, and APP/PS1 × $PSD-95^{+/-}$. The mice at 8 weeks of age were intraperitoneally injected with KA at a dosage of 30 mg/kg. We used a higher dosage of KA here to avoid a potential floor effect where seizure activity is too low in $PSD-95^{+/-}$, which would prevent accurate comparison between the four groups of mice. We used the modified Racine scoring scale to assess seizure activity as we did for Fig. 1. As shown in Fig. 7B, although genetically knocking down PSD-95 did not reduce the seizure susceptibility in APP/PS1 mice (APP/PS1 × $PSD-95^{+/-}$), it significantly reduced the lethality of mice following seizures. When we compared seizure activity between APP/PS1 and APP/PS1 × $PSD-95^{+/-}$ mice (Fig. 7C), we also observed significant reductions in the highest score and elevations of latency to stage 4 seizures in APP/PS1 × $PSD-95^{+/-}$ mice in comparison to APP/PS1 mice. WT and $PSD-95^{+/-}$ mice were excluded from these comparisons because most of them did not show stage 4 seizures (Fig. 7B). The attempt to conduct the same experiment using $PSD-95^{-/-}$ mice was not successful because APP/PS1 × $PSD-95^{-/-}$ mouse was not produced following a breeding scheme that crosses APP/PS1 × $PSD-95^{+/-}$ to APP/PS1 × $PSD-95^{+/-}$ mice, suggesting severe developmental defects in APP/PS1 × $PSD-95^{-/-}$ mice.

To rule out the possibility that knocking down PSD-95 prevents the mice from responding to kainic acid or even developing seizures, we employed an even higher dosage of kainic acid (45 mg/kg) in WT and $PSD-95^{+/-}$ mice. As shown in Fig. 7D, both WT and $PSD-95^{+/-}$ mice developed strong seizures with no significant differences in seizure susceptibility, lethality, highest score, or latency to stage 4 seizures between them. These data suggest that knocking down PSD-95 does not interfere with the ability of mice to respond to kainic acid-induced seizures. Instead, it significantly reduces seizure activity in young APP/PS1 mice. Because we observed elevated synapse numbers following $A\beta_{1-42}$ treatment in cultures (Fig. 5), we asked whether knocking down

PSD-95 affects excitatory synaptic transmission in APP/PS1 mice. To this end, we measured miniature excitatory postsynaptic currents (mEPSCs) of CA1 neurons in hippocampal slices from mice of four genotypes (WT, $PSD-95^{+/-}$, APP/PS1, and APP/PS1 × $PSD-95^{+/-}$). We chose the hippocampus because of its critical connection to epilepsy seizures in AD (Tombini et al, 2021). As shown (Fig. EV4), however, we did not observe significant changes in either the amplitude or frequency of mEPSCs across four genotypes. We suspect that the effects of elevated PSD-95 on synaptic transmission may be more profound in certain brain regions other than the hippocampal CA1 region. This would require additional investigation to validate in the future.

## Discussion

Our study has uncovered the mechanism by which an elevation of PSD-95 promotes seizure response in early $A\beta$ pathology (Fig. 7E). At the cellular level, we illustrated the roles of PSD-95 in $A\beta$-induced elevation of excitatory synapses and surface expression of AMPA receptors. Our findings show that the PSD-95-dependent alternations can be observed when APP/PS1 mice are only 8 weeks old, an age at which most of the disease phenotypes, such as plaque formation and memory dysfunction, have not yet developed in these mice. This suggests that $A\beta$-associated neuronal abnormalities can occur even before the onset of traditionally defined early $A\beta$ pathology. Therefore, our results introduce PSD-95 as a potential therapeutic target, especially during the early stage of the disease, to reduce brain hyperactivity. PSD-95 is known to exert multiple indispensable functions in the brain, including targeting and stabilization of ion channels and receptors, neural network maturation and stability, as well as synaptic integrity and plasticity (Araki et al, 2020; Chen et al, 2011; Yusifov et al, 2021). Furthermore, changes in the levels of PSD-95 are associated with alterations of behavior, including social interaction, anxiety, learning and memory (Stein et al, 2003; Vega-Torres et al, 2022; Winkler et al, 2018). Hence, it is undoubtedly critical to characterize the degree and duration of PSD-95 elevation during $A\beta$ pathology in order to minimize adverse effects following inhibition of PSD-95.

Several studies have observed reduced levels of PSD-95 in aged AD mouse models, which has led to the hypothesis that downregulation of PSD-95 contributes to impaired excitatory synaptic transmission in AD (Shao et al, 2011; Sultana et al, 2010). However, others have observed elevated PSD-95 during different stages of AD in animal models or patient samples (Dore et al, 2021; Kivisäkk et al, 2022; Leuba et al, 2008). This discrepancy suggests a possibility that PSD-95 may only be elevated in early stages of the disease. In fact, multiple clinical studies have observed that seizures and brain hyperexcitability occur more often in patients with early-onset disease, particularly when there is a familial presenilin I (PS1) mutation or abnormal expression of amyloid precursor protein (APP) (Vossel et al, 2017, 2013). Because we observed elevated PSD-95 in young APP/PS1 mice (8-week-old) and cultured treated with $A\beta$ peptide for only 2 h, these data suggested the possibility that PSD-95 may go from being elevated to being downregulated as the $A\beta$ pathology progresses into the later stage. As a valuable future direction, it would be important to characterize whether a transition of PSD-95 level occurs in APP/PS1 mice or other animal

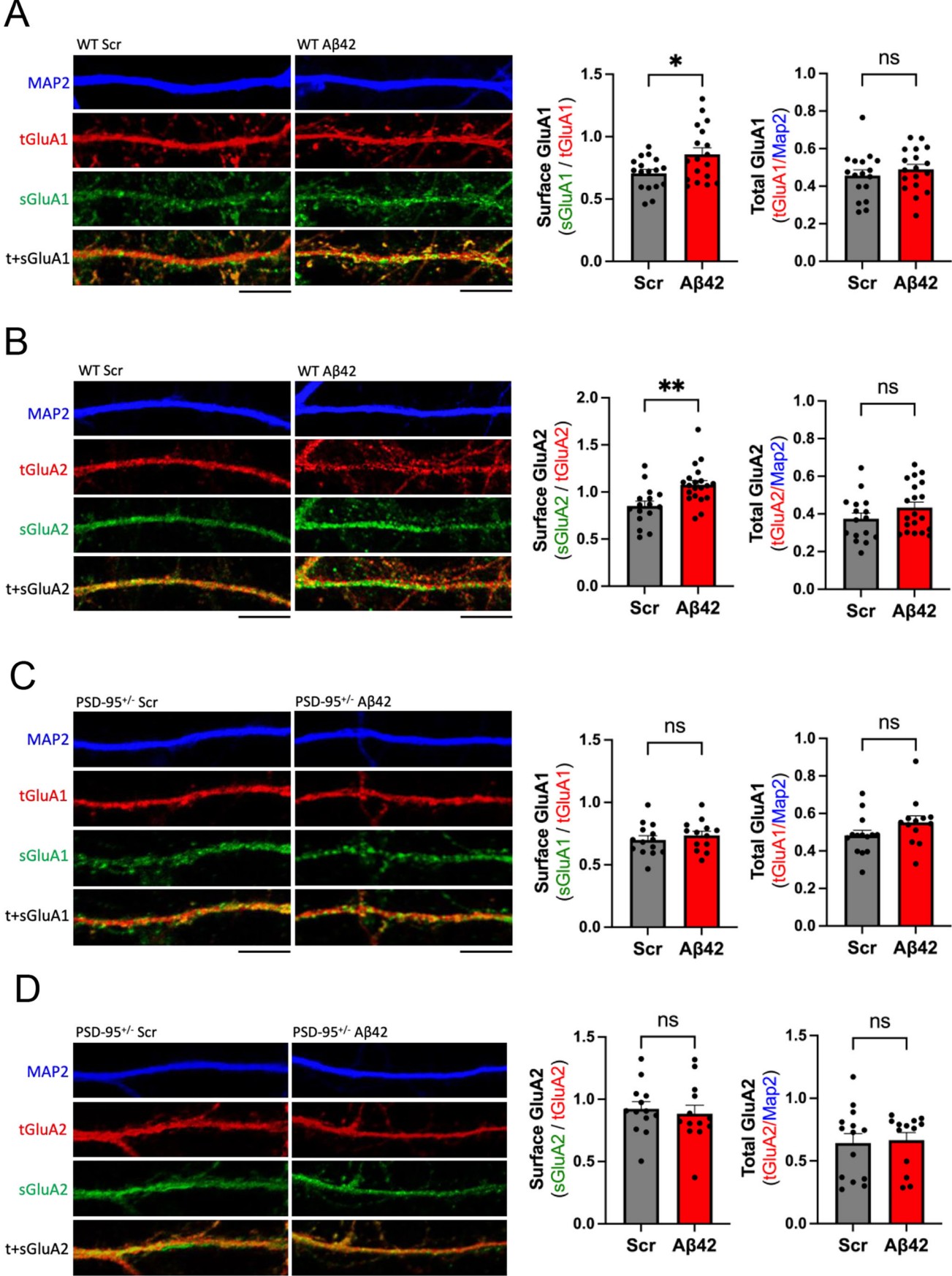

◄ **Figure 6. Aβ promotes PSD-95-dependent surface expression of AMPA receptors.**

(A, B) Immunocytochemistry and quantification showing dendritic marker MAP2, total (t) and surface (s) GluA1 (A) and GluA2 (B) from WT primary cortical neuron cultures treated with Aβ$_{1-42}$ (Aβ42, 1 μM) or scrambled peptide (Scr, 1 μM) for 2 h at DIV 12–14. For GluA1, *$P$ = 0.0192 for surface level, $P$ = 0.42 for total level, and $n$ = 17 and 18 cells from two independent cultures treated with Scr and Aβ42, respectively. For GluA2, **$P$ = 0.0025 for surface level, $P$ = 0.1584 for total level, and $n$ = 16 and 20 cells from two independent cultures treated with Scr and Aβ42, respectively. (C, D) Immunocytochemistry and quantification showing dendritic marker MAP2, total (t) and surface (s) GluA1 (C) and GluA2 (D) from PSD-95$^{+/-}$ primary cortical neuron cultures treated with Aβ$_{1-42}$ (Aβ42, 1 μM) or scrambled peptide (Scr, 1 μM) for 2 h at DIV 12–14. For GluA1, $P$ = 0.4539 for surface level, $P$ = 0.1311 for total level, and $n$ = 14 and 13 cells from two independent cultures treated with Scr and Aβ42, respectively. For GluA2, $P$ = 0.6643 for surface level, $P$ = 0.659 for total level, and $n$ = 13 and 14 cells from two independent cultures treated with Scr and Aβ42, respectively. Data Information: significance was determined by Student's $t$ test (A, C) or Mann–Whitney $U$ test (B, D). Data are represented as mean ± SEM with *$P$ < 0.05, **$P$ < 0.01, ns non-significant. Scale bar: 5 μm. Source data are available online for this figure.

models of Aβ pathology. If such a transition occurs, it would then be interesting to examine whether and how Mdm2 activity is also modulated accordingly. Furthermore, although we did not observe sex-dependent differences in seizure activity in young APP/PS1 mice, other studies have observed enhanced seizure behaviors in female APP/PS1 mice at an older age (Minkeviciene et al, 2009; Reyes-Marin and Nuñez, 2017). This information indicates the possibility that PSD-95 may be differentially regulated between male and female APP/PS1 mice when they age. This would need to be confirmed by future experiments.

Our previous work has shown that the levels of Mdm2 can be elevated when the cells receive unfavorable pathological stimuli that lead to the endoplasmic reticulum (ER) stress (Liu et al, 2019). Chronic accumulation of Aβ is known to activate ER stress pathways (Ajoolabady et al, 2022). This suggests that Mdm2-dependent PSD-95 ubiquitination may be elevated in the later phase of Aβ pathology when ER stress response is elevated, leading to down-regulation of PSD-95. Furthermore, one of our previous studies has shown that fragile X messenger ribonucleoprotein (FMRP) is required for sustaining the activity of Mdm2 under ER stress (Liu et al, 2021), while our another study showed that FMRP is required for chronic accumulation of Aβ-induced neural network hyposynchronicity (Lizarazo et al, 2022). Because others have shown that FMRP regulates the mRNA translation of PSD-95 (DeMarco et al, 2019), together with our studies, they suggest a possibility that FMRP regulates brain activity and PSD-95 expression over the course of Aβ pathology. Several recent studies have started to tie in FMRP with neurodegeneration (Wang, 2015), so it would be of great interest to understand how FMRP is regulated and participates at different stages of Aβ pathology.

In addition to potentiate AMPA receptors, an elevation of PSD-95 could directly or indirectly affect neural activity through other mechanisms. For example, the ratio of excitatory-to-inhibitory synaptic contacts may be altered following the changes in PSD-95 expression (Prange et al, 2004). It remains to be determined whether an elevation of PSD-95 is associated with reduced number of inhibitory synapses. Another mechanism underlying brain hyper-activity in Aβ pathology is the accumulation of glutamate due to impaired glutamate reuptake (Zott et al, 2019) or elevated release probability of presynaptic vesicles (Abramov et al, 2009; Russell et al, 2012). These observations support the hypothesis that the over-activated N-methyl-D-aspartate (NMDA) receptor, a glutamate receptor, contributes to the elevated excitotoxicity commonly seen in AD (Findley et al, 2019; Hascup et al, 2019). However, these studies do not explain why acute Aβ treatment can activate NMDA receptors even when glutamate is tightly controlled in healthy animals (You et al, 2012). While some studies have suggested that Aβ may bind and

activate NMDARs, such a scenario does not occur with the most toxic form of Aβ, the Aβ$_{1-42}$ (Danysz and Parsons, 2012). Collectively, this information led to our prediction that Aβ may exert other effects that potentiate NMDA receptors, perhaps through promoting the integrity of PSD. The integrity of PSD is positively correlated with the activity of NMDA receptors (Won et al, 2016). Because we observed an elevation of PSD-95, the key scaffolding component in PSD, there is a possibility that the hyperactivity of NMDA receptors in Aβ pathology is contributed in part by elevated PSD-95. Although we did not observe a significant elevation of NMDA receptors in PSD fractions of young APP/PS1 mice, it is still possible that other effectors or auxiliary proteins of NMDA receptors are potentiated in APP/PS1 mice, leading to hyperactivation of NMDA receptors. For example, in the PSD of young APP/PS1 mice, we observed elevated levels of Ca$^{2+}$/calmodulin-dependent protein kinase II-d (CaMKII-d; Fig. 4D), one of the major effector proteins downstream of NMDA receptors (Cook et al, 2022). Another elevated protein in the PSD of young APP/PS1 mice is neuroplastin (Nptn; Fig. 4D), which was reported to regulate calcium signaling downstream of NMDA receptors (Malci et al, 2022). Elevation of these proteins supports the likelihood that NMDA receptors as well as NMDA receptor-dependent excitotoxicity could appear early in Aβ pathology and that inhibition of PSD-95 may have a positive impact on ameliorating the excitotoxicity. Validating this prediction may further strengthen our original hypothesis that PSD-95-dependent neuronal defects occur early in the disease and that inhibition of PSD-95, at least during the early phase of the disease, may slow down the progression of the symptoms. We plan to test this idea in the future.

## Methods

### Ethics statement

This study followed the guidelines of Animal Care and Use provided by the University of Illinois Institutional Animal Care and Use Committee (IACUC) and the guidelines of Euthanasia of Animals provided by the American Veterinary Medical Association (AVMA) to minimize the number of animals used and animal suffering. This study was conducted under an approved IACUC animal protocol at the University of Illinois at Urbana-Champaign (#20049 and #23016 to N-PT).

### Animals

Wild-type (WT) (JAX 000664), APP/PS1 double-transgenic mice (JAX 005864, also MMRRC_034832-JAX) and PSD-95 null mice

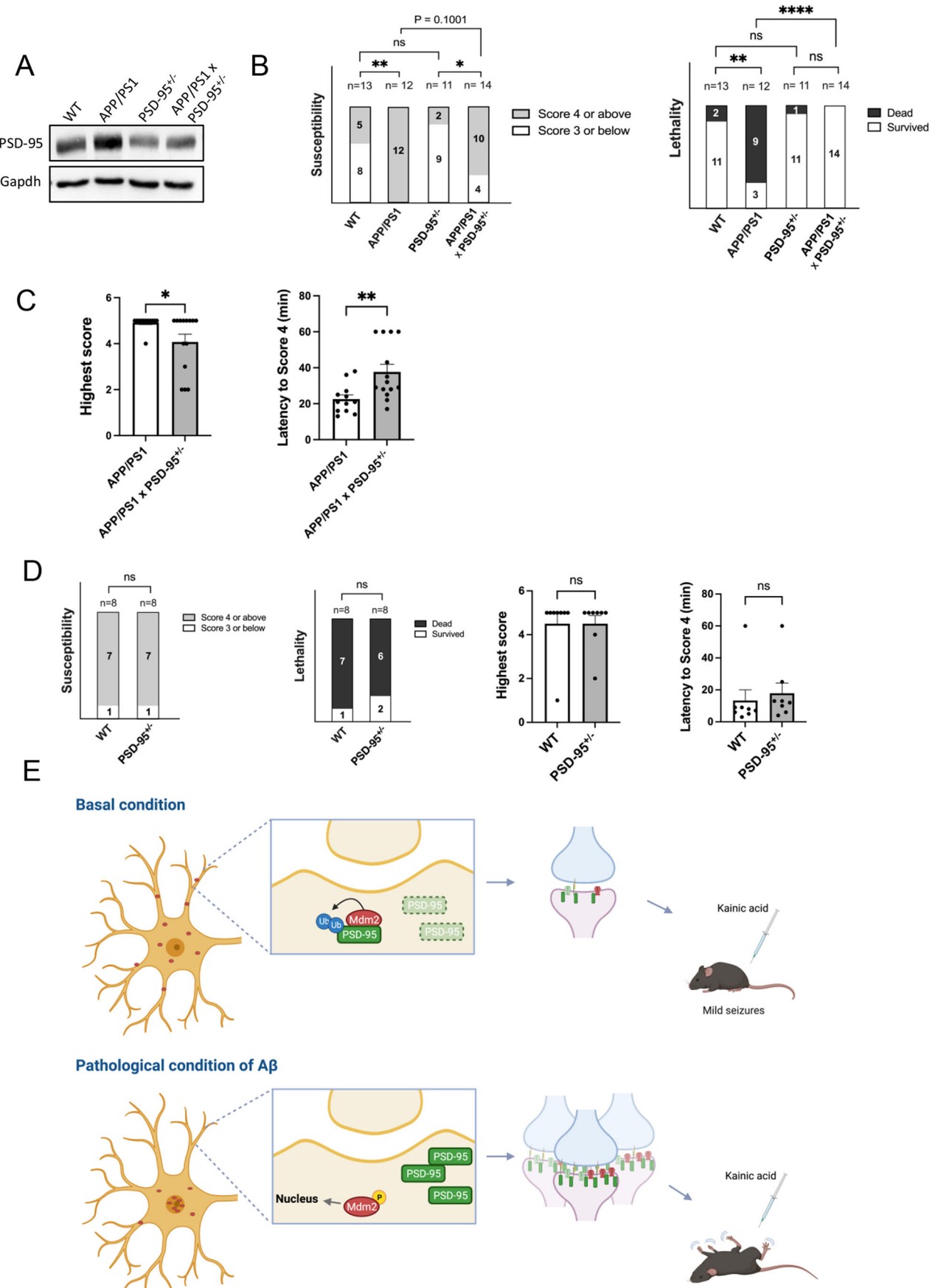

Figure 7. **PSD-95 suppression in APP/PS1 mice significantly reduces seizure severity.**

(A) Representative western blots of PSD-95 and Gapdh from the brain lysates of a group of WT, PSD-95$^{+/-}$, APP/PS1, and APP/PS1 × PSD-95$^{+/-}$ littermate mice.
(B) Quantification of susceptibility to stage 4 seizures and lethality from 8-weeks-old WT, PSD-95$^{+/-}$, APP/PS1, and APP/PS1 × PSD-95$^{+/-}$ mice intraperitoneally injected with kainic acid (30 mg/kg). For seizure susceptibility, **$P = 0.0016$ and *$P = 0.0154$. For lethality, **$P = 0.0048$ and ****$P < 0.0001$. $n = 11$–$14$ mice as indicated.
(C) Quantification of highest seizure scores (*$P = 0.0496$) and latency to stage 4 seizures (**$P = 0.0031$) from 8-weeks-old APP/PS1, and APP/PS1 × PSD-95$^{+/-}$ mice intraperitoneally injected with kainic acid (30 mg/kg). $n = 12$ and $14$ for APP/PS1, and APP/PS1 × PSD-95$^{+/-}$ mice, respectively. Sixty minutes in latency to score 4 means the mice without showing score 4 or above. (D) Quantification of susceptibility to stage 4 seizures ($P > 0.9999$), latency to stage 4 seizures ($P > 0.9999$), highest seizure scores ($P > 0.9999$), and lethality ($P > 0.9999$) from 8-weeks-old WT and PSD-95$^{+/-}$ mice intraperitoneally injected with kainic acid (45 mg/kg). $n = 8$ mice for both WT and PSD-95$^{+/-}$ mice. (E) A working model illustrating Mdm2-mediated ubiquitination of PSD-95 reduces seizure susceptibility while impairment of that leads to elevated seizure susceptibility during early Aβ pathology. Data Information: significance was determined by Fisher's exact test (seizure susceptibility and lethality) or Mann–Whitney $U$ test (highest score and latency to score 4). Data are represented as mean ± SEM with *$P < 0.05$ and **$P < 0.01$, ****$P < 0.0001$, ns non-significant. Source data are available online for this figure.

(JAX 013099) were backcrossed in C57BL/6J background. Mice were group-housed on a 12-h light/dark cycle in a temperature-controlled room with ad libitum feeding. APP/PS1 mice and PSD-95 null mice were crossed with WT mice by trio-breeding strategy and heterozygous mice were used throughout the study. For genotyping, the primers to detect WT and mutant alleles for APP/PS1 were: 5′-GTG TGA TCC ATT CCA TCA GC-3′ (WT forward), 5′-GGA TCT CTG AGG GGT CCA GT-3′ (Common), and 5′-ATG GTA GAG TAA GCG AGA ACA CG-3′ (Mutant forward). To detect WT and mutant alleles for PSD-95, the primers were: 5′-AAA CCC AGG AGC AGA GGT TTC ATG ACA CTA-3′, 5′-TCA TAG GGG TCC ATC AGT CTC TGC CT-3′, and 5′-ATG CTC CAC ACT GCC TTG GGA AAA G-3′. Polymerase chain reactions (PCR) for both mouse models were performed by the program as below: 2 min at 94 °C, 10 cycles of 20 s at 94 °C, 15 s at 65 °C (decreased by 0.5 °C per cycle), and 10 s at 68 °C; 28 cycles of 15 s at 94 °C, 15 s at 60 °C, and 10 s at 72 °C; and 2 min at 72 °C.

## Reagents

Dimethyl sulfoxide was from Thermo Fisher Scientific (#BP231). Kainic acid was from Cayman Chemical (#78050). Saline is from Hannas Pharmaceutical (#00409196612). Scrambled (#A-1004-1) and Aβ$_{1-42}$ (#A-1002-1) were synthesized by rPeptide. MK-2206 is from Selleckchem (#S1078). Rabbit anti-Mdm2 antibody (#A13327) and Rabbit anti-phospho-Mdm2-S166 antibody (#AP0073) are from ABClonal. Mouse anti-Gapdh antibody (#60004-1-lg) is from ProteinTech. Mouse anti-PSD-95 antibody (#sc32290), mouse anti-phospho-Mdm2-S166 antibody (#sc-965), and mouse anti-phospho-Mdm2-S186 antibody (#sc-53368) are from Santa Cruz Biotechnology. Rabbit anti-p53 antibody (#2527), rabbit anti-pan Akt antibody (#4691), rabbit anti-phospho-Akt-S473 antibody (#4060), rabbit anti-phospho Akt (T308) antibody (#13038), rabbit anti-ubiquitin antibody (#58395) and horseradish peroxidase (HRP)-conjugated anti-mouse secondary antibody (#7076) were from Cell Signaling Technology. Rabbit polyclonal anti-Synapsin-I antibody (ab8), chicken polyclonal anti-Microtubule-associated protein 2 antibody (MAP2, #ab92434), and rabbit anti-GluA1-C-terminus antibody (#Ab31232) are from Abcam. Rabbit anti-GluA2-C-terminus antibody (#SAB4501295) is from Sigma. Mouse anti-GluA1-N-terminus antibody (#MAB2263) and mouse anti-GluA2-N-terminus antibody (#MAB397) are from Millipore. Anti-mouse Homer1b/c antibody (#160111) is from Synaptic Systems. Anti-mouse Alexa Fluor 488 antibody (#A11001), anti-mouse Alexa Fluor 555 antibody (#A21422),

anti-rabbit Alexa Fluor 555 antibody (#A21428), anti-rabbit Alexa Fluor 633 antibody (#A2107) and anti-chicken Alexa Fluor 633 antibody (#A21103) were from Invitrogen. HRP-conjugated anti-rabbit secondary antibody (#711-035-152) was from Jackson ImmunoResearch.

## Behavioral seizure assessment

Mice at 8 weeks of age were intraperitoneally injected with kainic acid (KA) prepared in saline (Hannas Pharmaceutical) at the dosages indicated in each experiment to induce seizures. After injection, mice were monitored for an hour in real time. Modified Racine's scoring system was used to assess the seizure intensity (Fig. 1A).

## Surgical procedures for peptide injection

Mice were anesthetized continuously with inhalation of isoflurane and placed onto a stereotaxic frame (David Kopf Instruments). Throughout the procedure, the isoflurane-based anesthesia was closely monitored and modified. A heating pad was placed under the mouse to maintain the body temperature at 37.5 °C. After the toe pinch test, analgesia (Carprofen, 5 mg/kg) was given sub-cutaneously. A pre-pulled glass pipette (1B100-4; World Precision Instruments) filled with peptides (Scrambled or Aβ$_{1-42}$) dissolved in 10% DMSO + 90% PBS was placed into the lateral ventricle (AP + 0.3 from bregma, ML ± 1.0 from the midline, DV − 2.0 from dura), and peptides were unilaterally microinjected at 1 μl/min using a microinjection syringe pump (UMP3; World Precision Instruments). The glass pipettes were withdrawn after 10 min to avoid possible backflow of the peptide, and the incised skin was cleaned and sutured with sterilized suture thread. Mice were allowed to recover in a clean cage with food and water containing amoxicillin (50 mg/ml) for 5 days. Subsequent behavioral seizure experiments were conducted 5–6 days after the surgeries.

## Primary neuron cultures

Mixed-sex mice at postnatal (P) day 0–1 were used for primary neuron cultures. Cortices were collected, digested with trypsin, and triturated in Dulbecco's Modified Eagle Medium (DMEM) with DNase-I (1 mg/ml). Two to three hours after cells were placed on Poly-D-Lysin (PDL)-coated plates, the DMEM was replaced with NeuralA basal medium supplemented with B27 supplement, 2 mM GlutaMax, a mix of the antibiotics Penicillin (10,000 IU) and

Streptomycin (10,000 μg/mL), and 1 μM cytosine β-ᴅ-arabinofuranoside (AraC). Half of the culture medium was changed with new medium every 3–4 days until the cells were harvested at days-in-vitro (DIV) 12–14. Each experiment in this study was conducted with cultures made from at least two independent litters.

## Western blotting

Cortices from 8-week-old mice or cultured neurons at DIV 12–14 were isolated and homogenized in cold lysis buffer (137 mM NaCl, 2 mM EDTA, 1% Triton X-100, and 20 mM Tris-HCl, pH 8.0) with protease inhibitor (A32963, Thermo Fisher Scientific). Bradford assay was utilized to measure the protein concentration and protein samples were prepared in a sodium dodecyl sulfate (SDS) buffer (40% glycerol, 420 mM tris-HCL, pH 6.8, 8% SDS, 0.04% bromophenol blue, and 5% β-mercaptoethanol) at 0.5 μg/μl for brain lysates and at 1 μg/μl for cell lysates. Samples were loaded onto SDS-polyacrylamide gel electrophoresis (PAGE) gels, followed by gel electrophoresis. Proteins were transferred onto polyvinylidene fluoride (PVDF) membranes. The membranes were blocked by 1% bovine serum albumin (BSA) in Tris-buffered saline Tween-20 buffer (TBST; 20 mM Tris, pH 7.5, 150 mM NaCl, 0.1% Tween-20) for 30 min at room temperature, and then incubated overnight with primary antibodies at 4 °C. On the following day, the membranes were washed with TBST three times for 10 min each and then incubated with secondary antibodies in 5% milk in TBST for 1 h at room temperature. After washing the membranes with TBST for 10 min three times, the image was developed using iBright FL 1500 (Invitrogen).

## Immunocytochemistry and confocal microscopy

Primary neuron cultures were made from mixed-sex mice at P0 or P1 and subsequently maintained as described above on PDL-coated glass coverslips at a density of $1.2 \times 10^5$ cells/coverslip. Lentiviral transduction was performed on DIV 9 (Origene; PSD-95 lentiviral particle #RC215178L4V; control lentiviral particle #PS100093V). For the synapse counting experiment, coverslips were prepared as described previously (Lodes et al, 2022). Briefly, cells were washed with phosphate-buffered saline (PBS) once, fixed with fixation buffer (4% paraformaldehyde and 4% sucrose in PBS) for 15 min, and then permeabilized with permeabilization buffer (0.2% Triton X-100 in PBS) for 10 min. After that, the cells were blocked with 1% BSA in PBS for 30 min and incubated overnight with primary antibody at 4 °C. For WT and PSD-95$^{+/-}$ cultures, Synapsin-I, PSD-95, and MAP2 were used as a presynaptic marker, a postsynaptic marker, and a dendritic marker, respectively. For PSD-95$^{-/-}$ cultures, Synapsin-I and Homer1b/c were used as the presynaptic marker and the postsynaptic marker, respectively. On the following day, the neurons were washed three times for 10 min with PBS and incubated with fluorescence-conjugated secondary antibodies for 2 h at room temperature. Goat anti-mouse IgG Alexa Fluor 488, goat anti-rabbit IgG Alexa Fluor 555, and goat anti-chicken IgG Alexa Fluor 633 were used for WT and PSD-95$^{+/-}$ cultures, and goat anti-mouse IgG Alexa Fluor 555 and goat anti-rabbit IgG Alexa Fluor 633 were used for PSD-95$^{-/-}$ cultures. Following post-washing an additional three times for 10 min with PBS, the coverslips were mounted onto glass slides.

To assess the surface expression of GluA1 and GluA2, after one-time wash with PBS, cells on coverslips were fixed using fixation buffer as above for 8 min, washed two times for 5 min with PBS, blocked with 1% BSA in PBS for 1 h at room temperature, and incubated overnight with primary antibodies against N-terminus of GluA1 or GluA2 at 4 °C. The next day, cells were washed three times for 10 min with PBS, incubated with fluorescence-conjugated secondary antibodies (goat anti-mouse IgG Alexa Fluor 488), and washed three times additionally. To detect the C-terminus of GluA1 or GluA2, cells were fixed again for 20 min, washed three times for 10 min with PBS, and incubated for 10 min with permeabilization buffer. Following two times washes and blocking for 1 h as described above, cells were incubated overnight with primary antibodies against the C-terminal of subunits and MAP2. After overnight incubation, cells were washed three times for 10 min, incubated with secondary antibodies (goat anti-rabbit IgG Alexa Fluor 555, goat anti-chicken IgG Alexa Fluor 633), washed three times, and mounted onto glass slides.

Images were collected using a Zeiss LSM 700 confocal microscope with ×40 magnification. To differentiate three different protein markers, three laser lines (488, 555, and 633 nm) were utilized. The scanning configurations for the confocal microscope, such as the pinhole size and the laser intensity, were maintained for each experiment. To count the number of synapses, Image J with Neuron J and SynapCountJ plugins were used for analysis (Mata et al, 2017).

## Postsynaptic density enrichment and proteomics

Enrichment of proteins at postsynaptic density was performed following a modified method from Jang et al (Jang et al, 2017). Cortices from 8-week-old mice were homogenized with homogenization buffer (320 mM sucrose, 5 mM sodium pyrophosphate, 1 mM EDTA, 10 mM HEPES (pH 7.4), 200 nM Okadaic acid, Halt protease inhibitor). Homogenates were centrifuged at 800 × g for 10 min at 4 °C and the supernatant (S1) was collected and centrifuged at 13,800 × g for 10 min at 4 °C. Following the centrifugation, the pellet (P2) was resuspended with 4 mM HEPES, including protease inhibitors, and incubated with agitation for 30 min at 4 °C. Resuspended pellet (P2) was centrifuged at 25,000 × g for 20 min at 4 °C. After the centrifugation, the pellet (P3) was collected, resuspended with 25 mM HEPES, 0.5% Triton X-100 in PBS, and Halt protease inhibitor, and then incubated with agitation for 30 min at 4 °C. Following the incubation, the resuspended solution was centrifuged at 25,000 × g for 3 h at 4 °C. The final pellet was collected as PSD fractions. The proteomic profiling of PSD fractions was conducted by Creative Proteomics Inc. In brief, protein concentration was determined by BCA assay, and 100 μg protein per sample was precipitated using methanol and chloroform. After that, the protein was dissolved in 2 M urea solution, denatured, and digested with Trypsin at 37 °C for 15 h. To remove salt, the peptides were purified with C18 SPE columns and prepared in 0.1% formic acid. Ultra-performance liquid chromatography-tandem mass spectrometry (UPLS-MC/MC) was performed using the Ultimate 3000 nano UHPLC system (Thermo Scientific). The raw data files were analyzed and searched against the mouse protein database using Maxquant (1.6.2.14).

## Whole-cell patch-clamp recordings

Mice (6–8 weeks of age) were decapitated after anesthetizing with isoflurane. The brains were rapidly removed and placed in an ice-cold dissection solution containing (in mM) 220 sucrose, 2.5 KCl, 1 $Na_2PO_4$, 2.5 $MgCl_2$, 0.5 $CaCl_2$, 25 $NaHCO_3$, and 20 D-glucose, bubbled with 95% $O_2$ and 5% $CO_2$. Coronal slices (350-μm thick) were prepared using a vibratome (VT 1000S; Leica, Germany). Hippocampal slices were collected in oxygenated artificial cerebrospinal fluid (aCSF) containing (in mM) 119 NaCl, 2.5 KCl, 1 $Na_2PO_4$, 1.3 $MgCl_2$, 2.5 $CaCl_2$, 26 $NaHCO_3$, and 20 D-glucose, bubbled with 95% $O_2$ and 5% $CO_2$. The slices were incubated at aCSF for 1 hr at 32 °C prior to all recordings. Whole-cell patch-clamp recordings were performed as previously described (Lee et al, 2023). In brief, the recordings were performed on hippocampal CA1 pyramidal neurons at 30–32 °C in a submersion chamber continuously perfused with aCSF. For miniature excitatory postsynaptic currents (mEPSCs), bicuculine (20 μM), and TTX (0.5 μM) were added to the aCSF to block $GABA_A$ receptors and sodium channel receptors, respectively. Whole-cell recording pipettes (~3–5 MΩ) were filled with intracellular solution containing (in mM) 130 K-gluconate, 6 KCl, 3 NaCl, 10 HEPES, 0.2 EGTA, 4 Mg-ATP, 0.4 Na-GTP, 14 Tris-phosphocreatine, 2 QX-314 (pH 7.25, 285 mOsm). Membrane potential was clamped at −70 mV. Neurons were not included in analyses if the resting membrane potential was >−50 mV, access resistance was >30 MΩ, or if access resistance changed by >20%. All recordings were performed with Clampex 10.6 and Multiclamp 700B amplifier interfaced with Digidata 1550B data acquisition system (Molecular Devices). Recordings were filtered at 1 kHz and digitized at 10 kHz.

## Experimental design and statistical analysis

To minimize the variability, sister cultures made from the same litter of at least three pups were prepared at the same time. The cell suspensions were pooled before plating. For western blotting, at least three different litters were used for each experiment. For seizure experiments, at least three liters of littermates were used for each condition in each experiment. Additionally, litters from APP/PS1 mice crossed with PSD-95$^{+/−}$ mice for seizure experiments were conducted blindly without knowing the genotypes beforehand. For immunocytochemistry, at least two liters were used for each experiment. Expected effect sizes and sample sizes for experiments were decided based on our previous studies (Liu et al, 2019; Lizarazo et al, 2022; Lodes et al, 2022; Zhu et al, 2019). No samples or animals were excluded from our analyses. The data presented in this study have been tested for normality using the Shapiro–Wilk normality test and then analyzed by parametric or nonparametric tests. When two conditions were compared, the two-tailed Student's $t$ test or Mann–Whitney test was used depending on the normality of each data. For multiple comparisons between genotypes, the Kruskal–Wallis test and Dunn's multiple comparisons test were conducted. For seizure experiments, seizure susceptibility and lethality were analyzed by Fisher's exact test. Outlier was determined by Grubb's outlier test. Statistical analysis was performed using GraphPad 9. Error bars represent SEM with *$P < 0.05$, **$P < 0.01$, ***$P < 0.001$, and ****$P < 0.0001$.

## Data availability

The proteomic profiling results were deposited at figshare.com under the file name "Proteomic results from PSD fractions of WT or APPPS1 mice" and can be accessed via https://doi.org/10.6084/m9.figshare.24904716.

## Peer review information

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

## Acknowledgements

The authors would like to thank Sophia Azim and the Imaging Facility at the Institute of Genomic Biology at the University of Illinois Urbana-Champaign for the technical help. This work is supported by the National Institute of Health R01NS105615, R01MH124827, and R21AG071278 as well as Alzheimer's Association AARG-23-1149282 to N-PT.

## Author contributions

**Yeeun Yook**: Conceptualization; Data curation; Formal analysis; Investigation; Methodology; Writing—original draft; Writing—review and editing. **Kwan Young Lee**: Data curation; Formal analysis; Investigation. **Eunyoung Kim**: Data curation; Investigation. **Simon Lizarazo**: Data curation; Formal analysis. **Xinzhu Yu**: Resources; Supervision. **Nien-Pei Tsai**: Conceptualization; Resources; Supervision; Funding acquisition; Investigation; Methodology; Writing—original draft; Project administration; Writing—review and editing.

## Disclosure and competing interests statement

The authors declare no competing interests.

# Expanded View Figures

**A**  KA 15mg/kg, APP/PS1

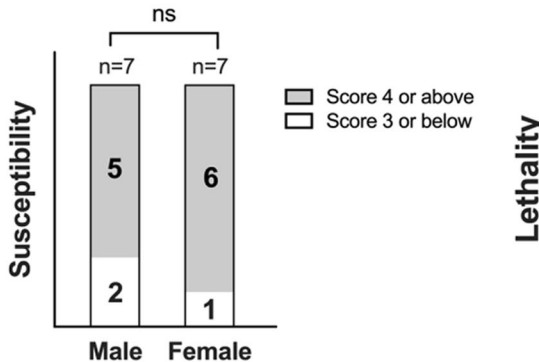
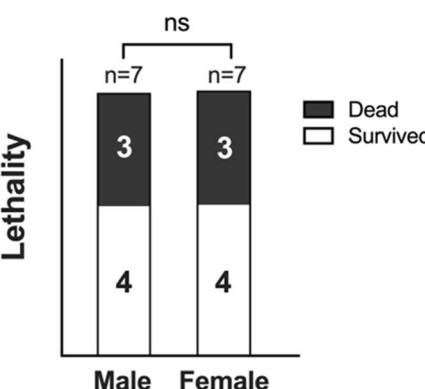
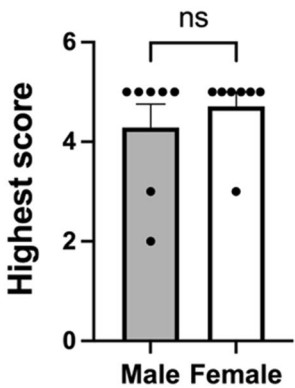

**B**  KA 30mg/kg, APP/PS1

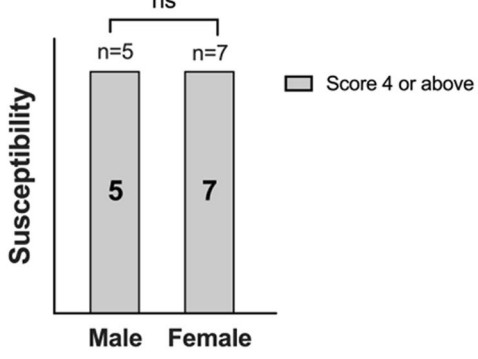
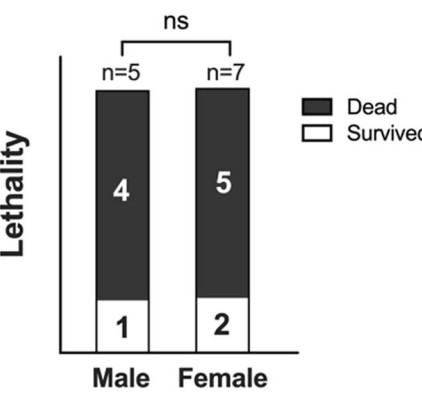
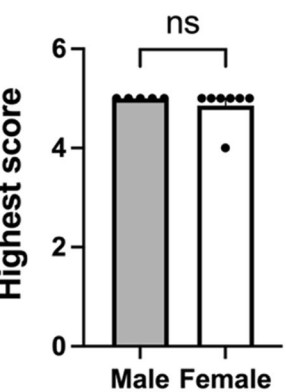

**Figure EV1.  Seizure severity in young APP/PS1 does not exhibit sex differences.**

(A, B) Quantification of seizure susceptibility, lethality, and highest score in male and female APP/PS1 mice. Two different doses of 15 mg/kg (A) and 30 mg/kg (B) were intraperitoneally injected into mice. Significance was determined by Fisher's exact test (seizure susceptibility; $P > 0.9999$ for both 15 and 30 mg/kg and lethality; $P > 0.9999$ for both 15 and 30 mg/kg) or Mann–Whitney $U$ test (highest score; $P = 0.7308$ for 15 mg/kg and $P > 0.9999$ for 30 mg/kg). Data are represented as mean ± SEM with ns: non-significant. Source data are available online for this figure.

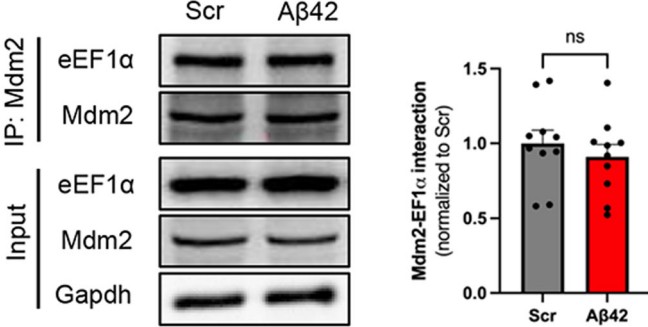

**Figure EV2. Aβ₁₋₄₂ does not alter the interaction between eEF1α and Mdm2.**

Quantification of interaction between eEF1α and Mdm2 after co-immunoprecipitation and representative western blots using lysates from WT primary cortical neuron cultures treated with Aβ₁₋₄₂ (Aβ42, 1 μM) or scrambled Aβ peptide (Scr, 1 μM) for 2 h at DIV 12–14. $n = 10$ at least from three independent cultures for both Scr and Aβ42 groups. Significance was determined by Student's $t$ test ($P = 0.4605$). Data are represented as mean ± SEM with ns: non-significant. Source data are available online for this figure.

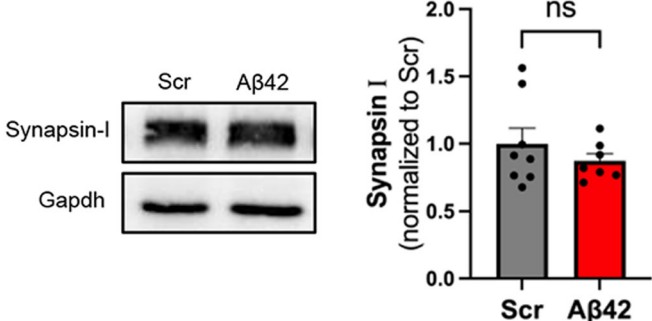

**Figure EV3.  Aβ<sub>1-42</sub> does not promote the total level of Synapsin-I.**

Quantification of Synapsin-I and representative western blots from WT primary cortical neuron cultures treated with amyloid-beta 1–42 (Aβ42; 1 µM) or scrambled Aβ peptide (Scr, 1 µM) for 2 h at DIV 12–14. $n = 8$ and 7 for Scr and Aβ42 groups, respectively. Significance was determined by Student's $t$ test ($P = 0.3604$). Data are represented as mean ± SEM with ns: non-significant. Source data are available online for this figure.

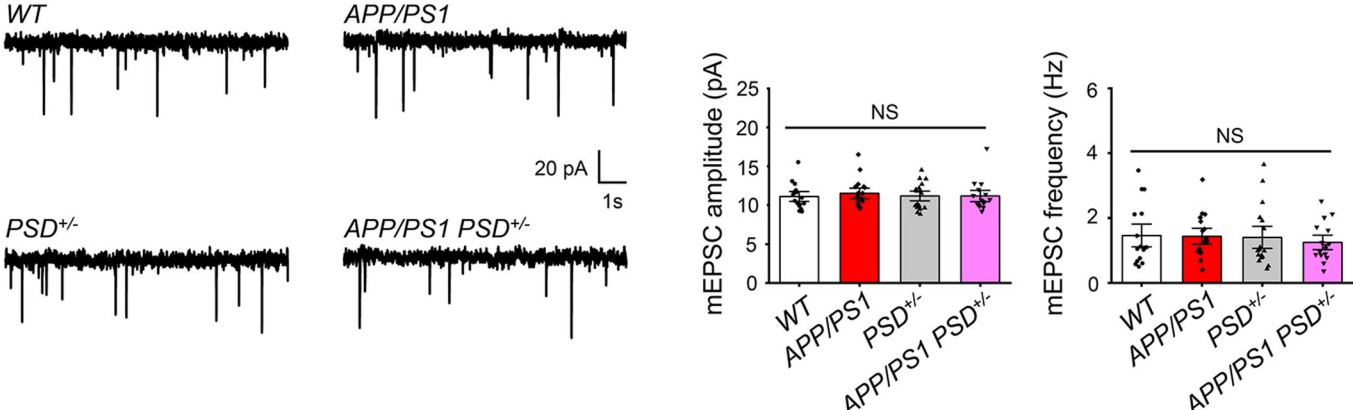

**Figure EV4. No changes in mEPSCs were observed in hippocampal CA1 neurons in PSD-95⁻/⁺ or APP/PS1 mice.**

Voltage-clamp recordings of mEPSCs from CA1 pyramidal cells in acute hippocampal slice of WT ($n = 16$ cells from 6 mice), APP/PS1 ($n = 17$ cells from 7 mice), PSD$^{-/+}$ ($n = 17$ cells from 6 mice), and APP/PS1 PSD$^{-/+}$ (16 cells from 6 mice) mice. Holding potential was −70 mV. Representative mEPSC traces (left) and quantification of mEPSC amplitude and frequency (right) are shown. Data were analyzed by one-way ANOVA with Tukey test and presented as mean ± SEM with NS: non-significant. Source data are available online for this figure.

