## [Peer Review File · EMBO Reports]

Hyperfunction of Post-synaptic Density Protein 95 Promotes Seizure Response in Early-Stage A β Pathology

Yeeun Yook, Kwan Young Lee, Eunyoung Kim, Simon Lizarazo, Xinzhu Yu, and Nien-Pei Tsai

Corresponding author(s): Nien-Pei Tsai (nptsai@illinois.edu)

Review Timeline:

Submission Date:	30th May 23
Editorial Decision:	12th Jul 23
Revision Received:	27th Dec 23
Editorial Decision:	24th Jan 24
Revision Received:	24th Jan 24
Accepted:	30th Jan 24

Editor: Esther Schnapp

Transaction Report:

Dear Nien-Pei,

Thank you for the submission of your manuscript to EMBO reports. We have now received the full set of referee reports that is pasted below.

As you will see, the referees acknowledge that the findings are potentially interesting. However, they also raise several concerns and have several suggestions for how the study should be strengthened. I think all comments are valid and should be addressed, according to what we discussed by email.

I would thus like to invite you to revise your manuscript with the understanding that the referee concerns must be fully addressed and their suggestions taken on board. Please address all referee concerns in a complete point-by-point response. Acceptance of the manuscript will depend on a positive outcome of a second round of review. It is EMBO reports policy to allow a single round of major revision only and acceptance or rejection of the manuscript will therefore depend on the completeness of your responses included in the next, final version of the manuscript.

We realize that it is difficult to revise to a specific deadline. In the interest of protecting the conceptual advance provided by the work, we recommend a revision within 3 months (12th Oct 2023). Please discuss the revision progress ahead of this time with the editor if you require more time to complete the revisions.

- 1) A data availability section providing access to data deposited in public databases is missing. If you have not deposited any data, please add a sentence to the data availability section that explains that.
- 2) Your manuscript contains statistics and error bars based on $n=2$. Please use scatter blots in these cases. No statistics should be calculated if $n=2$.

5) a complete author checklist, which you can download from our author guidelines <https://www.embopress.org/page/journal/14693178/authorguide>. Please insert information in the checklist that is also reflected in the manuscript. The completed author checklist will also be part of the RPF.

6) Please note that all corresponding authors are required to supply an ORCID ID for their name upon submission of a revised manuscript (<https://orcid.org/>). Please find instructions on how to link your ORCID ID to your account in our manuscript tracking system in our Author guidelines <https://www.embopress.org/page/journal/14693178/authorguide#authorshipguidelines>

7) Before submitting your revision, primary datasets produced in this study need to be deposited in an appropriate public

database (see <https://www.embopress.org/page/journal/14693178/authorguide#datadeposition>). Please remember to provide a reviewer password if the datasets are not yet public. The accession numbers and database should be listed in a formal "Data Availability" section placed after Materials & Method (see also <https://www.embopress.org/page/journal/14693178/authorguide#datadeposition>). Please note that the Data Availability Section is restricted to new primary data that are part of this study. * Note - All links should resolve to a page where the data can be accessed. *

10) Regarding data quantification (see Figure Legends:

<https://www.embopress.org/page/journal/14693178/authorguide#figureformat>)

12) All Materials and Methods need to be described in the main text. We would encourage you to use 'Structured Methods', our new Materials and Methods format. According to this format, the Materials and Methods section should include a Reagents and Tools Table (listing key reagents, experimental models, software and relevant equipment and including their sources and relevant identifiers) followed by a Methods and Protocols section in which we encourage the authors to describe their methods using a step-by-step protocol format with bullet points, to facilitate the adoption of the methodologies across labs. More information on how to adhere to this format as well as downloadable templates (.doc or .xls) for the Reagents and Tools Table can be found in our author guidelines under 'structured methods':

<<https://www.embopress.org/page/journal/17444292/authorguide#textformat>>. An example of a Method paper with Structured Methods can be found here: <<https://www.embopress.org/doi/10.15252/msb.20178071>>.

You are able to opt out of this by letting the editorial office know (emboreports@embo.org). If you do opt out, the Review Process File link will point to the following statement: "No Review Process File is available with this article, as the authors have

chosen not to make the review process public in this case."

I look forward to seeing a revised form of your manuscript when it is ready.

Referee #1:

To summarize, they first show that young (8 weeks old) APP/PS1 mice are more susceptible to seizure induction than WT mice. In exploring the molecular underpinnings of this effect, they show increased phosphorylation of Mdm2 (a ubiquitin ligase) in both APP/PS1 mice and cultured neurons treated with AB42. When non-phosphorylated, Mdm2 is known to ubiquitinate PSD-95 and promote its degradation, so they show the knock-on effect of elevated PSD-95 expression and reduced ubiquitination in APP/PS1 mice and AB42 treated cultures. They then do proteomics on the isolated PSD fraction of cortical homogenates from WT and APP/PS1 mice and show a predominant elevation of synaptic proteins in the APP/PS1 mice. To verify if this leads to increased synapse numbers, they compare stainings of WT neuronal cultures treated for 2 hours with AB42 or scrambled control peptide and find that AB induces an increase in synapse numbers and surface expression of AMPA receptors. This effect was not seen in cultures prepared from PSD95^{+/-} mice. Finally, they test whether seizure susceptibility is decreased in APP/PS1xPSD95^{+/-} mice and show that although the crosses have a similar susceptibility to seizures as the APP/PS1 mice, the seizures result in significantly lower lethality. They, therefore, propose that elevated AB acts to promote the phosphorylation of Mdm2, which in turn leads to elevated PSD-95 levels, resulting in increased synapse density and neuronal excitability, causing increased seizure susceptibility.

I have several concerns:

- I do not find it convincing that they have used PSD95^{+/-} mice for their experiments as it creates some ambiguity for the seizure susceptibility results in Figure 6. Does the lack of change to seizure susceptibility from APP/PS1s result from there still being residual PSD95 expression or is it due to alternative mechanisms that they have not accounted for? If they had used a full PSD95 knockout combined with APP/PS1 I would predict it would result in a more convincing rescue of the seizure phenotype if their proposed mechanism were true.
- Similar to the above, I would find the following experiment to be a more convincing means to show that AB increases synapse numbers through PSD-95. Prepare cultures from PSD95^{-/-} mice instead of PSD95^{+/-} and reintroduce PSD95 using a viral vector. Then compare the response of PSD95^{-/-} and PSD^{-/-} + virus cultures to AB42.
- The observation of elevated PSD-95 in APP/PS1 mice and after administration of AB42 somewhat contrasts with available literature in similar mouse models and human tissue (see, e.g., Dore et al, Cell Reports, 2021) and needs some discussion
- In Figure 5A, they have normalized their synaptic counts to control levels. I am not sure why this is necessary. The data can just be stated as absolute densities.
- There is an assumption that the phosphorylation status of Mdm2 impacts its activity, but this is not fully addressed or referenced
- Amounts of AB42 peptides used in this study is very high (1 μ M) and it would be more convincing to see if effects hold with nanomolar concentrations
- Line 163-164: "or a control scrambled peptide..." - authors should clarify what peptide was used here
- There is an assumption that Akt is a major kinase that phosphorylates Mdm2 at S166 but that is not addressed or properly referenced
- Line 172-173: "These data suggest that AB-induced Mdm2 phosphorylation is likely independent of Akt." This conclusion appears to be an assumption rather than a definitive outcome of the experiments. Just because there was no observed change in Akt phosphorylation, it does not conclusively imply that AB-induced Mdm2 phosphorylation is independent of Akt. Furthermore, a lack of changes in EF1 α -Mdm2 interaction following AB42 treatment further clouds the understanding of the mechanism behind A β -induced Mdm2 phosphorylation. A more comprehensive exploration of potential mechanisms, beyond

just stating an 'uncharacterized mechanism', would strengthen the study.

- In Figure 4, while normalizing to PSD-95 levels seems reasonable given its role and observed elevation in APP/PS1 mice, the authors should provide further justification for why PSD-95 is the best candidate for normalization and discuss any potential limitations if PSD-95 also changes in a way unrelated to synaptic strength, as this could confound the analysis.

- While the reduced seizure activity in PSD-95^{+/-} mice even after administration of a higher dosage of kainic acid provides some compelling evidence, it would be important to clarify whether the different dosages might have potentially induced seizures via different mechanisms in the two mouse strains. Also, the authors did not directly demonstrate that the observed reduction in seizures is a result of altered excitatory synaptic transmission, which is the central thesis of their paper. They could consider additional electrophysiological experiments to substantiate their claim.

- The authors need to sort out the format of their references before it can be published (in some papers all the authors are just given as initials)

Referee #2:

In the present manuscript Yook et al. tackled a very important question in the field of Alzheimer's disease (AD). Both familial and sporadic forms of this neurodegenerative disorder present neuronal network hyperactivity in specific phases (usually in the early steps) of disease progression, which, in some cases evolve to epileptic seizures. Yet, the cellular mechanisms and molecular actors underlying these events remain to be clarified. The authors herein elucidate a putative molecular pathway via hyperfunction of PSD-95 in cortical glutamatergic neurons.

Regarding the data I have some concerns that should be addressed as follows:

1. The authors recurrently mention "Abeta pathology in an earlier stage". I guess the authors base this on the fact they are using the APP/PS1 model at 8 weeks-old. The authors should clarify their "early stage criteria" to allow understanding the impact of the study. The author's hypothesis is that Abeta triggers Mdm2-P interfering with PSD-95, thus causing its increase. The weakness of this hypothesis is the use of an AD model that presents increased APP beta-processing, leading to the production of several APP fragments, including C99 and AICD, which are known to also affect glutamatergic synapses (see review Müller et al. 2017). Considering this, authors should better characterize their model at 8 weeks: is Abeta increased in cortical tissue? in which form? Do animals present Abeta plaques? Is it different between females and males? Also, a better way to establish a causality between Abeta and seizures susceptibility would be to inject Abeta species (monomers and/or oligomers) directly in the cortex of adult WT animals and evaluate if this causes higher susceptibility to Kainic acid 15mg/kg induced seizures.

2. In lines 123 - 126, the authors claim their study provide the first demonstration of the molecular mechanisms underlying elevated seizure susceptibility during the early stage of Abeta pathology. I do think that the data presented by authors is insufficient. Abeta - seizures susceptibility causality is not proven, and how Abeta induced Mdm2-P remains unknown - as authors mention "through an uncharacterized mechanism" (line 180, Figure 2).

3. In Figure 1 authors evaluated seizures susceptibility with KA at a dose of 15 mg/Kg. From 16 WT animals, 6 presented seizures with score 4 or above. In Figure 6 authors used KA 30 mg/Kg and 2 animals on 10 got score 4. This means that with higher dose of KA WT animals present lower seizures susceptibility. How do authors explain this observation? Moreover, in the title, authors state "hyperfunction of PSD-95 promotes seizure susceptibility in early-stage Abeta pathology", but in Figure 6 seizure susceptibility was not rescued by crossing APP/PS1 with the PSD95 heterozygous (Figure 6B). As authors mention (line 272) PSD-95 suppression significantly reduces seizure severity in APP/PS1 mice.

4. Figure 3E - H is hard to understand as authors do not provide the quantitative methods analysis of the western blots. Also, this results section (Lines 185 - 200 should be rephrased to better understand the data.

5. Seizures result from an inhibitory / excitatory balance and are often associated to a gabaergic loss of function. In this case authors suggest that in AD, seizures susceptibility is triggered by increased PSD-95 / AMPA expression in cortical neurons. The remaining question is the functional output of this increase. The article would highly benefit from the electrophysiological study on brain slices from APP/PS1 PSD 95 ^{+/-} in order to access excitatory / inhibitory balance.

Referee #3:

Overall rating:

Novel but need Major Revisions

Suitability for publication:

The paper is potentially suitable for publication but requires major revisions and additional experiments.

Clarity/figures:

The clarity of the paper is good but the figures need corrections and clarifications.

1. Figure 2A should include the full membrane as there seems to have additional bands has been cropped out that are not explained.
2. Figure 4C needs a terminology correction. Please correct Gria1 to GluA1 as here is refer to protein instead of gene, and also consistent with the descriptions in line 226.
3. Figure 5C-D could be strengthened with additional experiments.

The paper "Hyperfunction of PSD-95 Promotes Seizure Susceptibility in Early-Stage A β Pathology" provides novel insights into the role of PSD-95 in the early stages of Alzheimer's disease (AD). This perspective is significant; however, there are several areas where the paper could be strengthened:

1. In Figure 2A, it is critical to include the full membrane as it seems there are additional bands above and below has been cropped that are not explained.
2. Regarding Mdm2 phosphorylation, it is not clear why only examine S166, please clarify whether other phosphorylation sites of Mdm2, such as Serine 186 (Ser186), Serine 17 (Ser17), and Serine 395 (Ser395) have been considered. Especially Ser186 need to be examined, as it has the similar effect as Ser 166 to enhance Mdm2's ability to ubiquitinate and target p53 for degradation.
3. On page 9, line 193, the claim "Specifically, when phosphorylated, Mdm2 interacts less with PSD-95, leading to reduced ubiquitination and degradation of PSD-95" needs to be substantiated. Though the cited paper (Colledge et al., 2003) showed Mdm2 is required for ubiquitination and degradation of PSD-95, but they did not specifically mention or investigate how phosphorylation of Mdm2 impacts this process in this paper.
4. In Figure 4C, please change the term "Gria1" to "GluA1" as you are referring to the protein, not the gene.
5. For Figure 5C-D, to support the claim in line 251 that "PSD-95 is required for A β 1-42-induced elevation of excitatory synapses in cortical neurons", it would be more convincing to include experiments using PSD-95 knockout neurons.
6. For Figure 6, please consider experiments examining if PSD-95 knockout in APP/PS1 mice further reduce lethality following seizures, and whether this effect is gene-dosage dependent.
7. While the paper proposes that PSD-95 inhibition may have therapeutic potential for AD, it is important to investigate the hypersocial behavior in PSD95 \pm mice, as demonstrated by Winkler et al., Behavioural Brain Research, 2018. Moreover, discuss or examine how the current findings align with previous studies that have highlighted the importance of PSD-95 in neural network stability, molecular organization of the postsynaptic density, and behavior (Winkler et al., Behavioural Brain Research, 2018; Chen et al., Journal of Neuroscience, 2011; Yusifov et al., Proceedings of the National Academy of Sciences, 2021).

We would like to thank the reviewers for their constructive comments on our manuscript. As we outlined below, we have added more data points in our experiments (new Fig. 7B, previous Fig. 6B), provided a substantial amount of new data (Figs. 1D, parts of 2A-2E, 5C-5D, and EV4) and carefully revised the text suggested by the reviewers. The changes in manuscript are marked in red. We believe the manuscript is now greatly improved as a consequence of the reviewers' suggestions. In the following, we have provided point-by-point responses to reviewers' comments in BLUE.

Referee 1:

To summarize, they first show that young (8 weeks old) APP/PS1 mice are more susceptible to seizure induction than WTs. In exploring the molecular underpinnings of this effect, they show increased phosphorylation of Mdm2 (a ubiquitin ligase) in both APP/PS1 mice and cultured neurons treated with AB42. When non-phosphorylated, Mdm2 is known to ubiquitinate PSD-95 and promote its degradation, so they show the knock-on effect of elevated PSD-95 expression and reduced ubiquitination in APP/PS1 mice and AB42 treated cultures. They then do proteomics on the isolated PSD fraction of cortical homogenates from WT and APP/PS1 mice and show a predominant elevation of synaptic proteins in the APP/PS1 mice. To verify if this leads to increased synapse numbers, they compare stainings of WT neuronal cultures treated for 2 hours with AB42 or scrambled control peptide and find that AB induces an increase in synapse numbers and surface expression of AMPA receptors. This effect was not seen in cultures prepared from PSD95^{+/-} mice. Finally, they test whether seizure susceptibility is decreased in APP/PS1^xPSD95^{+/-} mice and show that although the crosses have a similar susceptibility to seizures as the APP/PS1 mice, the seizures result in significantly lower lethality. They, therefore, propose that elevated AB acts to promote the phosphorylation of Mdm2, which in turn leads to elevated PSD-95 levels, resulting in increased synapse density and neuronal excitability, causing increased seizure susceptibility.

I have several concerns:

- I do not find it convincing that they have used PSD95^{+/-} mice for their experiments as it creates some ambiguity for the seizure susceptibility results in Figure 6. Does the lack of change to seizure susceptibility from APP/PS1s result from there still being residual PSD95 expression or is it due to alternative mechanisms that they have not accounted for? If they had used a full PSD95 knockout combined with APP/PS1 I would predict it would result in a more convincing rescue of the seizure phenotype if their proposed mechanism were true.

To address this comment, we crossed PSD-95^{+/-} × APP/PS1 with PSD-95^{+/-} × APP/PS1 in order to obtain PSD-95^{-/-} × APP/PS1. However, among 105 pups that were born over the period of four months, we were only able to obtain one PSD-95^{-/-} × APP/PS1 mouse, and this mouse is extremely runty. These observations suggest that PSD-95^{-/-} × APP/PS1 likely have severe developmental deficits. We have described our observation in page 15 in the manuscript. Due to this limitation, we were unable to perform seizure experiments using these mice. However, we

were able to employ PSD-95^{-/-} (KO) neurons to address the next comment raised by the reviewer.

- Similar to the above, I would find the following experiment to be a more convincing means to show that AB increases synapse numbers through PSD-95. Prepare cultures from PSD95^{-/-} mice instead of PSD95^{+/-} and reintroduce PSD95 using a viral vector. Then compare the response of PSD95^{-/-} and PSD^{-/-} + virus cultures to AB42.

As suggested by the reviewer, we prepared cultures from PSD-95^{-/-} mice and reintroduce *PSD-95* using lentivirus. As shown in the new Figs. 5C and 5D, A β -induced elevation of synapse numbers is inhibited PSD-95^{-/-} cultures but can be restored following lentiviral transduction of *PSD-95*. Please note that we used Homer1b/c as the post-synaptic marker in this experiment since PSD-95 is not present. Together, these results support our conclusion that PSD-95 is crucial to A β -induced elevation in synapse numbers.

- The observation of elevated PSD-95 in APP/PS1 mice and after administration of AB42 somewhat contrasts with available literature in similar mouse models and human tissue (see, e.g., Dore et al, Cell Reports, 2021) and needs some discussion

Dore et al. and others have found that chronic treatment of A β (18-24 hours) or older APP/PS1 mice (> 6 months of age) exhibit reduced PSD-95 levels and a decrease in synapse numbers, which were also observed in our previous study (Lizarazo et al., 2022). In our current study, we observed the effects of elevated PSD-95 following A β treatment for only 2 hours as well as in young APP/PS1 mice (8 weeks of age). These data suggest that the hyperactivity resulted from elevated PSD-95 likely occurs during the early stage of A β pathology before the neurons enter into hypoactivity that is potentially contributed by reduced PSD-95 during the later stage of A β pathology. We have discussion regarding this matter in page 17 of the manuscript.

- In Figure 5A, they have normalized their synaptic counts to control levels. I am not sure why this is necessary. The data can just be stated as absolute densities.

We removed the normalization process. The data are now presented as absolute value (puncta per μ m dendrite).

- There is an assumption that the phosphorylation status of Mdm2 impacts its activity, but this is not fully addressed or referenced

We have now referenced three papers regarding the link between Mdm2 phosphorylation and its activity in page 8 of the manuscript.

- Amounts of AB42 peptides used in this study is very high (1 μ M) and it would be more convincing to see if effects hold with nanomolar concentrations

To address this comment, we treated primary cortical neurons with 100 nM A β 42 peptide and tested the levels of Mdm2 phosphorylation and PSD-95 (please see below). While the phosphorylation of Mdm2 is slightly increased, the effect was not significant ($p=0.1543$) and PSD-95 levels were not changed. These results suggest that acute A β 42 peptide treatment (2 hours) at 100 nM is not sufficient to promote PSD-95 elevation. This observation is in fact consistent with two previous studies (Brorson et al., 1995; Ciccone et al., 2019) showing neuronal hyperactivity following A β peptide treatment in the range of 1 to 10 μ M. We therefore believe 1 μ M is ideal to induce neuronal hyperactivity in our cultured neurons. We have included these two citations and the justification of choosing 1 μ M in page 9 of the manuscript.

- Line 163-164: "or a control scrambled peptide..." - authors should clarify what peptide was used here

The scrambled peptide is made of the same amino acid constituents of A β 42 but with randomized sequence. This peptide was employed as the control treatment throughout our study. We have revised the main text and figure legends to ensure consistency.

- There is an assumption that Akt is a major kinase that phosphorylates Mdm2 at S166 but that is not addressed or properly referenced

We have included additional references in page 9 to emphasize the known roles of Akt in Mdm2 phosphorylation.

- Line 172-173: "These data suggest that AB-induced Mdm2 phosphorylation is likely independent of Akt." This conclusion appears to be an assumption rather than a definitive outcome of the experiments. Just because there was no observed change in Akt phosphorylation, it does not conclusively imply that AB-induced Mdm2 phosphorylation is independent of Akt. Furthermore, a lack of changes in EF1 α -Mdm2 interaction following AB42 treatment further clouds the understanding of the mechanism behind A β -induced Mdm2 phosphorylation. A more comprehensive exploration of potential mechanisms, beyond just stating an 'uncharacterized mechanism', would strengthen the study.

We thank the reviewer for bringing up this point. In the revision, we employed an Akt inhibitor MK-2206 and confirmed that Akt is indeed required for A β -induced phosphorylation of Mdm2 (new Fig. 2C). In addition to Akt phosphorylation at S473 that we tested in our first submission, we tested another Akt phosphorylation site T308 in this revision (Figs. 2D and 2E). However, we

did not detect changes in Akt phosphorylation at either of these sites following A β treatment. These data suggest that Mdm2 phosphorylation is mediated by Akt but is likely independent of changes in Akt phosphorylation at these two known residues.

- In Figure 4, while normalizing to PSD-95 levels seems reasonable given its role and observed elevation in APP/PS1 mice, the authors should provide further justification for why PSD-95 is the best candidate for normalization and discuss any potential limitations if PSD-95 also changes in a way unrelated to synaptic strength, as this could confound the analysis.

We have described our justification and the limitation in pages 11-12 of the manuscript. In brief, we chose PSD-95 because there is no information thus far indicating which protein might be unchanged in PSD of APP/PS1 mice. Instead of choosing a random protein, we chose PSD-95 that is shown to be elevated in APP/PS1 mice based on our data. Although there is a limitation that any proteins whose elevation is smaller than that of PSD-95 may be overlooked, normalization through PSD-95 can point out those proteins that are being further enriched in the PSD fractions of APP/PS1 mice, even beyond the already enriched PSD-95.

- While the reduced seizure activity in PSD-95^{+/-} mice even after administration of a higher dosage of kainic acid provides some compelling evidence, it would be important to clarify whether the different dosages might have potentially induced seizures via different mechanisms in the two mouse strains. Also, the authors did not directly demonstrate that the observed reduction in seizures is a result of altered excitatory synaptic transmission, which is the central thesis of their paper. They could consider additional electrophysiological experiments to substantiate their claim.

The two strains of mice (PSD-95^{+/-} mice and APP/PS1 mice) are both in the same genetic backgrounds (C57BL/6J), and all genetic groups of mice used in our seizure experiments are littermates. It is unlikely that seizures are induced via different mechanisms in these mice at different dosages of kainic acid.

Reviewer 2 also suggested electrophysiology experiment and recommended using brain slices. In this revision, we have performed mEPSC recording in CA1 neurons in hippocampal slices of all four groups (WT, APP/PS1, PSD-95^{+/-} and APP/PS1 PSD-95^{+/-}) to assess basal synaptic transmission. We chose hippocampus because of its critical connection to epilepsy seizures in AD (Tombini et al., 2021). As shown in Figure EV4, however, we did not observe significant difference across four genotypes. We suspect that the effects of elevated PSD-95 on synaptic transmission may be more profound in certain brain regions other than hippocampal CA1 region. This will require further investigation to validate in the future. We have described these new data and the need for a future direction in page 16 of the manuscript.

- The authors need to sort out the format of their references before it can be published (in some papers all the authors are just given as initials)

We have revised the format of references. Thank you for pointing out this issue.

Referee 2:

In the present manuscript Yook et al. tackled a very important question in the field of Alzheimer's disease (AD). Both familial and sporadic forms of this neurodegenerative disorder present neuronal network hyperactivity in specific phases (usually in the early steps) of disease progression, which, in some cases evolve to epileptic seizures. Yet, the cellular mechanisms and molecular actors underlying these events remain to be clarified. The authors herein elucidate a putative molecular pathway via hyperfunction of PSD-95 in cortical glutamatergic neurons.

Regarding the data I have some concerns that should be addressed as follows:

1. The authors recurrently mention "Abeta pathology in an earlier stage". I guess the authors base this on the fact they are using the APP/PS1 model at 8 weeks-old. The authors should clarify their "early stage criteria" to allow understanding the impact of the study.

We defined the early stage as when mice start to accumulate plasma A β but without detectable plaques or defects in memory behavior based on a previous study (He et al., 2013). We have included this information in page 6 of the manuscript.

The author's hypothesis is that Abeta triggers Mdm2-P interfering with PSD-95, thus causing its increase. The weakness of this hypothesis is the use of an AD model that presents increased APP beta-processing, leading to the production of several APP fragments, including C99 and AICD, which are known to also affect glutamatergic synapses (see review Müller et al. 2017). Considering this, authors should better characterize their model at 8 weeks: is Abeta increased in cortical tissue? in which form? Do animals present Abeta plaques? Is it different between females and males?

These APP/PS1 mice have been extensively characterized by other studies. In brief, A β 42 (approximately 450pg/ml) can be detected in these mice at 8 weeks of age (He et al., 2013) but amyloid plaques only begin to emerge in the cortex at 4 months of age and in the hippocampus at 6 months (Jackson et al., 2013; Minkeviciene et al., 2008). Sex-dependent changes typically happens in aged APP/PS1 mice; females exhibited a higher concentration of plasma A β than males at 15 months (Ordóñez-Gutiérrez et al., 2015). Because this information is already available in literatures, we therefore did not characterize them in our revision. We hope the reviewer would understand. Instead, we have provided the information along with these references in page 7 of the manuscript.

Also, a better way to establish a causality between Abeta and seizures susceptibility would be to inject Abeta species (monomers and/or oligomers) directly in the cortex of adult WT animals and evaluate if this causes higher susceptibility to Kainic acid 15mg/kg induced seizures.

As suggested by the reviewer, we performed a new experiment by stereotaxically injecting scrambled or A β 42 peptides into the ventricle of WT mice at 7 weeks old. Five to six days later, mice were intraperitoneally injected with kainic acid at 15 mg/kg. As shown (new Figs. 1D1 and

1D2), mice injected with A β 42 showed more severe seizures compared to mice injected with scrambled peptide. One thing to notice is that mice injected with A β 42 did not exhibit lethality following seizures as we observed in APP/PS1 mice, suggesting less overall burden than APP/PS1 mice. These results are now described in page 8 of the manuscript.

2. In lines 123 - 126, the authors claim their study provide the first demonstration of the molecular mechanisms underlying elevated seizure susceptibility during the early stage of Abeta pathology. I do think that the data presented by authors is insufficient. Abeta - seizures susceptibility causality is not proven, and how Abeta induced Mdm2-P remains unknown - as authors mention "through an uncharaterized mechanism" (line 180, Figure 2).

As described above, our new data in Figs. 1D1 and 1D2 have strengthened the causality between A β and seizures. To address another question raised by Reviewer 1, we have provided new data to confirm Akt-dependent Mdm2 phosphorylation by A β in new Fig. 2C. Together with our findings showing the necessity of PSD-95 following Mdm2 phosphorylation in elevated synapse numbers in cultures and seizure susceptibility in APP/PS1 mice, we hope the reviewer would agree with us that our data have sufficiently supported our claim about a new mechanism underlying A β -induced hyperactivity.

3. In Figure 1 authors evaluated seizures susceptibility with KA at a dose of 15 mg/Kg. From 16 WT animals, 6 presented seizures with score 4 or above. In Figure 6 authors used KA 30 mg/Kg and 2 animals on 10 got score 4. This means that with higher dose of KA WT animals present lower seizures susceptibility. How do authors explain this observation? Moreover, in the title, authors state "hyperfunction of PSD-95 promotes seizure susceptibility in early-stage Abeta pathology", but in Figure 6 seizure susceptibility was not rescued by crossing APP/PS1 with the PSD95 heterozygous (Figure 6B). As authors mention (line 272) PSD-95 suppression significantly reduces seizure severity in APP/PS1 mice.

We reason that the issue with seizure susceptibility in WT mice between Fig-1 and Fig-6 (now Fig-7) is in part due to the lower sample size in old Fig-6 (now Fig-7). In this revision, we have performed additional experiments to add more mice to the experiment presented in the new Fig-7. The newly updated data suggest more comparable seizure susceptibility in WT mice between the two figures.

Also, we changed the title to "Hyperfunction of PSD-95 promotes seizure response in early-stage A β pathology".

4. Figure 3E - H is hard to understand as authors do not provide the quatitative methods analysis of the western blots. Also, this results section (Lines 185 - 200 should be rephrased to better understand the data.

We have added additional information in pages 10-11 to elaborate on our analyses of these data.

5. Seizures result from an inhibitory / excitatory balance and are often associated to a gabaergic loss of function. In this case authors suggest that in AD, seizures susceptibility is triggered by increased PSD-95 / AMPA expression in cortical neurons. The remaining question is the functional output of this increase. The article would highly benefit from the electrophysiological study on brain slices from APP/PS1 PSD 95 +/- in order to access excitatory / inhibitory balance.

The electrophysiology experiment was also suggested by Reviewer 1. During the revision, we have performed mEPSC recording in CA1 neurons of hippocampal slices from all four groups (WT, APP/PS1, PSD-95^{+/-} and APP/PS1 PSD-95^{+/-}) to assess basal synaptic transmission. We chose hippocampus because of its critical connection to epilepsy seizures in APP/PS1 mice (Tombini et al., 2021). As shown in Figure EV4, however, we could not observe significant difference across four genotypes. We suspect that the effects of elevated PSD-95 on synaptic transmission may be more profound in certain brain regions other than hippocampal CA1 region. This will require further investigation in the future. We have described these new data and the need for a future direction in page 16 of the manuscript. We did not pursue the analysis of inhibitory transmission because of our focus on PSD and PSD-95. However, we agree with the reviewer that inhibitory input is also critical to understand hyperactivity in AD, and we therefore have acknowledged this as a potential future direction (pages 18-19 of the manuscript).

Referee 3:

Remarks to the Author:

The paper "Hyperfunction of PSD-95 Promotes Seizure Susceptibility in Early-Stage A β Pathology" provides novel insights into the role of PSD-95 in the early stages of Alzheimer's disease (AD). This perspective is significant; however, there are several areas where the paper could be strengthened:

1. In Figure 2A, it is critical to include the full membrane as it seems there are additional bands above and below has been cropped that are not explained.

In the revision, we have provided uncropped images for all western blots presented in our manuscript. Additional bands on Mdm2 blots were cropped out in Fig-2A as those bands are considered non-specific based on our previous study using Mdm2 knockout animals (Liu et al., 2019).

2. Regarding Mdm2 phosphorylation, it is not clear why only examine S166, please clarify whether other phosphorylation sites of Mdm2, such as Serine 186 (Ser186), Serine 17 (Ser17), and Serine 395 (Ser395) have been considered. Especially Ser186 need to be examined, as it has the similar effect as Ser 166 to enhance Mdm2's ability to ubiquitinate and target p53 for degradation.

We examined S166 because this is the residue in which its phosphorylation is linked to the ubiquitination of both p53 and PSD-95 (Liu et al., 2019; Tsai et al., 2016). As suggested, we have measured phosphorylation of S186 in our revision. As shown in Figs. 2A-2B, we did not observe significant changes in phosphorylation at S186 in the lysate of either brains of APP/PS1 mice or primary cultures treated with A β 42 peptide. Because our new data confirmed the necessity of Akt in A β -induced Mdm2 phosphorylation (new Fig. 2C), and S166 and S186 are the major sites phosphorylated by Akt (Mayo & Donner, 2001), we therefore did not pursue the analyses of other phosphorylation sites.

3. On page 9, line 193, the claim "Specifically, when phosphorylated, Mdm2 interacts less with PSD-95, leading to reduced ubiquitination and degradation of PSD-95" needs to be substantiated. Though the cited paper (Colledge et al., 2003) showed Mdm2 is required for ubiquitination and degradation of PSD-95, but they did not specifically mention or investigate how phosphorylation of Mdm2 impacts this process in this paper.

We apologize for the confusion. The correct references (Tsai et al., 2012, 2016) have been added in page 10. In these papers, which were published by us and our colleagues, we have shown that Mdm2 phosphorylation leads to reduced PSD-95 ubiquitination.

4. In Figure 4C, please change the term "Gria1" to "GluA1" as you are referring to the protein, not the gene.

We have revised gene names to protein names in Fig. 4C.

5. For Figure 5C-D, to support the claim in line 251 that "PSD-95 is required for A β 1-42-induced elevation of excitatory synapses in cortical neurons", it would be more convincing to include experiments using PSD-95 knockout neurons.

This was also suggested by Reviewer 1. In this revision, we have prepared primary cultures using PSD-95^{-/-} mice. As shown in Fig. 5C, A β -induced elevation of synapse numbers is abolished in PSD-95^{-/-} neurons. As suggested by Reviewer 1, we further showed that lentivirally re-introducing PSD-95 in PSD-95^{-/-} neurons can restore A β -induced elevation of synapse numbers (Fig. 5D). Please note that, because PSD-95 is knocked out, we used Homer1b/c as the postsynaptic marker in these new data. Together, these data from PSD-95^{-/-} neurons strengthen our claim that PSD-95 is required for A β -induced elevation of synapse numbers.

6. For Figure 6, please consider experiments examining if PSD-95 knockout in APP/PS1 mice further reduce lethality following seizures, and whether this effect is gene-dosage dependent.

This was also suggested by Reviewer 1. As we explained to Reviewer 1, we crossed PSD-95^{+/-} \times APP/PS1 with PSD-95^{+/-} \times APP/PS1 in order to obtain PSD-95^{-/-} \times APP/PS1. However, we obtained only one PSD-95^{-/-} APP/PS1 mouse out of 105 pups over a period of 4 months. This suggests that PSD-95^{-/-} \times APP/PS1 likely have severe developmental deficits. We have described

this observation in page 15 in the manuscript. Due to this technical limitation, we were unable to complete this task.

7. While the paper proposes that PSD-95 inhibition may have therapeutic potential for AD, it is important to investigate the hypersocial behavior in PSD95^{+/-} mice, as demonstrated by Winkler et al., *Behavioural Brain Research*, 2018. Moreover, discuss or examine how the current findings align with previous studies that have highlighted the importance of PSD-95 in neural network stability, molecular organization of the postsynaptic density, and behavior (Winkler et al., *Behavioural Brain Research*, 2018; Chen et al., *Journal of Neuroscience*, 2011; Yusifov et al., *Proceedings of the National Academy of Sciences*, 2021).

Because our focus is on seizure response, we believe social activity can be a future direction to investigate. However, we agree with the reviewer that it is important to discuss about other known functions of PSD-95 that might affect therapeutic potentials of PSD-95 inhibition. In the revision, we provided additional discussion to cover this matter in page 17.

Remarks to the Editor:

The paper provides a novel perspective on the role of PSD-95 in early-stage A β pathology in Alzheimer's disease. It is well-written and can be an important contribution to the field. However, major revisions are necessary for the paper to meet the scientific standards of EMBO reports. Specifically, the paper needs additional experiments, corrections to the figures to support their statement.

References

- Brorson, J. R., Bindokas, V. P., Iwama, T., Marcuccilli, C. J., Chisholm, J. C., & Miller, R. J. (1995). The Ca²⁺ influx induced by β -amyloid peptide 25-35 in cultured hippocampal neurons results from network excitation. *Journal of Neurobiology*, 26(3), 325–338. <https://doi.org/10.1002/neu.480260305>
- Ciccione, R., Franco, C., Piccialli, I., Boscia, F., Casamassa, A., de Rosa, V., Cepparulo, P., Cataldi, M., Annunziato, L., & Pannaccione, A. (2019). Amyloid β -Induced Upregulation of Nav1.6 Underlies Neuronal Hyperactivity in Tg2576 Alzheimer's Disease Mouse Model. *Scientific Reports*, 9(1), 13592. <https://doi.org/10.1038/s41598-019-50018-1>
- He, J., Qiao, J.-P., Zhu, S., Xue, M., Chen, W., Wang, X., Tempier, A., Huang, Q., Kong, J., & Li, X.-M. (2013). Serum-Amyloid Peptide Levels Spike in the Early Stage of Alzheimer- Like Plaque Pathology in an APP/PS1 Double Transgenic Mouse Model. *Current Alzheimer Research*, 10(9), 979–986. <https://doi.org/10.2174/15672050113106660159>
- Jackson, H. M., Soto, I., Graham, L. C., Carter, G. W., & Howell, G. R. (2013). Clustering of transcriptional profiles identifies changes to insulin signaling as an early event in a mouse model of Alzheimer's disease. *BMC Genomics*, 14(1), 831. <https://doi.org/10.1186/1471-2164-14-831>

- Liu, D. C., Eagleman, D. E., & Tsai, N. P. (2019). Novel roles of ER stress in repressing neural activity and seizures through Mdm2- and p53-dependent protein translation. *PLoS Genetics*, *15*(9), e1008364. <https://doi.org/10.1371/JOURNAL.PGEN.1008364>
- Lizarazo, S., Yook, Y., & Tsai, N. P. (2022). Amyloid beta induces Fmr1-dependent translational suppression and hyposynchrony of neural activity via phosphorylation of eIF2 α and eEF2. *Journal of Cellular Physiology*, *237*(7), 2929–2942. <https://doi.org/10.1002/JCP.30754>
- Mayo, L. D., & Donner, D. B. (2001). A phosphatidylinositol 3-kinase/Akt pathway promotes translocation of Mdm2 from the cytoplasm to the nucleus. *Proceedings of the National Academy of Sciences*, *98*(20), 11598–11603. <https://doi.org/10.1073/pnas.181181198>
- Minkeviciene, R., Ihalainen, J., Malm, T., Matilainen, O., Keksa-Goldsteine, V., Goldsteins, G., Iivonen, H., Leguit, N., Glennon, J., Koistinaho, J., Banerjee, P., & Tanila, H. (2008). Age-related decrease in stimulated glutamate release and vesicular glutamate transporters in APP/PS1 transgenic and wild-type mice. *Journal of Neurochemistry*, *105*(3), 584–594. <https://doi.org/10.1111/j.1471-4159.2007.05147.x>
- Ordóñez-Gutiérrez, L., Antón, M., & Wandosell, F. (2015). Peripheral Amyloid Levels Present Gender Differences Associated with Aging in A β PP/PS1 Mice. *Journal of Alzheimer's Disease*, *44*(4), 1063–1068. <https://doi.org/10.3233/JAD-141158>
- Tombini, M., Assenza, G., Ricci, L., Lanzone, J., Boscarino, M., Vico, C., Magliozzi, A., & Di Lazzaro, V. (2021). Temporal Lobe Epilepsy and Alzheimer's Disease: From Preclinical to Clinical Evidence of a Strong Association. *Journal of Alzheimer's Disease Reports*, *5*(1), 243–261. <https://doi.org/10.3233/ADR-200286>
- Tsai, N.-P., Wilkerson, J. R., Guo, W., & Huber, K. M. (2016). FMRP-dependent Mdm2 dephosphorylation is required for MEF2-induced synapse elimination. *Human Molecular Genetics*, *25*(12), ddw386. <https://doi.org/10.1093/hmg/ddw386>
- Tsai, N.-P., Wilkerson, J. R., Guo, W., Maksimova, M. A., DeMartino, G. N., Cowan, C. W., & Huber, K. M. (2012). Multiple Autism-Linked Genes Mediate Synapse Elimination via Proteasomal Degradation of a Synaptic Scaffold PSD-95. *Cell*, *151*(7), 1581–1594. <https://doi.org/10.1016/j.cell.2012.11.040>

Dear Dr. Tsai,

Thank you for the submission of your revised manuscript. We have now received the enclosed reports from the referees that were asked to assess it. I am happy to say that both support the publication of your study now. Only a few more minor editorial requests will need to be addressed before we can proceed with the official acceptance of your manuscript:

- Please move the Data Availability Section (DAS) to directly after the methods sections and before the Acknowl. section.
- Please rename the conflict of interest subheading to "Disclosure and Competing Interests Statement"
- The reference format needs to be corrected to the EMBO reports style (in EndNote): et al needs to be used after 10 author names, DOIs should be used only for preprints and datasets that have not been published yet, year should be in brackets
- Table EV1 needs to be corrected to Dataset - please rename and upload as Dataset EV1; the callout in the ms needs to be corrected too.
- Please group the source data for EV figures and upload them as one single zipped folder.
- Please correct "METHODS" to "MATERIALS AND METHODS"
- Please add the database name for the "doi.org/10.6084/m9.figshare.24904716" dataset to the data availability statement.
- A separate 'Data Information' section is required in the legends of figures 1c-d; 2a-e; 5a-d; 6a-d; 7a-d.
- Please indicate the statistical test used for data analysis in the legend of figure 4b.

I would like to suggest a few minor changes to the abstract. Please let me know whether you agree with :

Accumulation of amyloid-beta ($A\beta$) can lead to the formation of aggregates that contribute to neurodegeneration in Alzheimer's disease (AD). Despite globally reduced neural activity during AD onset, recent studies have suggested that $A\beta$ induces hyperexcitability and seizure-like activity during early stages of the disease that ultimately exacerbate cognitive decline. However, the underlying mechanism is unknown. Here, we reveal an $A\beta$ -induced elevation of postsynaptic density protein 95 (PSD-95) in cultured neurons in vitro and in an in vivo AD model using APP/PS1 mice at 8 weeks of age. Elevation of PSD-95 occurs as a result of reduced ubiquitination caused by Akt-dependent phosphorylation of E3 ubiquitin ligase murine-double-minute 2 (Mdm2). The elevation of PSD-95 is consistent with the facilitation of excitatory synapses and the surface expression of α -amino-3-hydroxy-5-methyl-4-isoxazole propionic acid (AMPA) receptors induced by $A\beta$. Inhibition of PSD-95 corrects these $A\beta$ -induced synaptic defects and reduces seizure activity in APP/PS1 mice. Our results demonstrate a mechanism underlying elevated seizure activity during early-stage $A\beta$ pathology and suggest that PSD-95 could be an early biomarker and novel therapeutic target for AD.

Regarding the short summary you sent, this is supposed to summarise the main findings of your study. At the moment, it only contains current knowledge. Please send us a new summary. The bullet points are fine.

Referee #1:

Authors have addressed my comments.

Referee #2:

We highly appreciated the added experiments in order to clarify our main concerns. The authors made a significant effort and the manuscript clearly benefit by this review process. I have no further comments to be addressed.

The authors have addressed all minor editorial requests.

Dr. Nien-Pei Tsai
University of Illinois at Urbana-Champaign
Molecular and Integrative Physiology
407 South Goodwin Ave
Urbana, IL 61801
United States

Dear Dr. Tsai,

I am very pleased to accept your manuscript for publication in the next available issue of EMBO reports. Thank you for your contribution to our journal.

Yours sincerely,
